# ECLayr: Fast and Robust Topological Layer based on Differentiable Euler Characteristic Curve

## Abstract

In the realm of Topological Data Analysis, persistent homology has traditionally served as a primary tool for extracting topological features. However, approaches relying on persistent homology often encounter practical challenges due to their high computational costs. To address this issue, we propose a computationally efficient novel topological layer tailored for general deep learning architectures, leveraging the Euler Characteristic Curve (ECC). Unlike methods based on persistent homology, ECC offers computational advantages by circumventing the need for persistent homology calculation, while still allowing access to crucial information about the underlying topological structure. The proposed layer can readily adapt to diverse data modalities by allowing appropriate filtration according to the user's preference, enabling its application across various learning problems without data preprocessing. We present a novel technique for stable backpropagation that effectively mitigates the vanishing gradient problems commonly encountered in existing methods, allowing for seamless integration of our layer into deep learning models. We go on to present stability analysis, showing that the proposed layer is robust against noise and outliers. We apply our method to topological autoencoders, showing that the standard loss function can effectively regularize topological structures of the latent space. Through classification experiments across various datasets, we illustrate the benefits of our approach in mitigating information loss under conditions of data scarcity or data contamination.

## 1 Introduction

In recent years, machine learning communities have witnessed increasing efforts to incorporate Topological Data Analysis (TDA) into deep learning workflows, an emerging paradigm known as topological deep learning (Carlsson & Gabrielsson, 2020; Papamarkou et al., 2024). Topological deep learning integrates tools from TDA to exploit essential topological features within the data that elude conventional methods, or to enhance understanding and control of computational models. *Persistent homology* (PH), a primary tool in TDA, captures multi-scale topological features of the underlying data structure by tracking the birth and death of homology features, thereby producing topological summaries such as persistence diagrams or barcodes (Chazal & Michel, 2021). Given that PH is a multiset by nature, a number of strategies have been proposed to transform these PH-based topological summaries into alternative representations that are more suitable for subsequent machine learning tasks (e.g., Bubenik et al., 2015; Adams et al., 2017; Umeda, 2017) (see, for example, Hensel et al. (2021) for a review).

Recent efforts have shed light on the possibility of incorporating PH-based topological summaries as input features for neural networks, enhancing their ability to learn from the intrinsic geometric structure of the data. Hofer et al. (2017); Carrière et al. (2020) introduced topological layers aimed at learning vector embeddings of persistence diagrams using a particular parametrization, yet lacking differentiability required for enabling gradient backpropagation. Hofer et al. (2019); Gabrielsson et al. (2020); Carriere et al. (2021); Leygonie et al. (2022) explored the differentiability aspects of the PH-based functions or losses, highlighting the potential of incorporating topological insights into deep learning frameworks. Kim et al. (2020) were the first to propose a generic differentiable

topological layer allowing backpropagation, offering flexibility in its integration within arbitrary network architectures.

Despite its popularity, computations involving PH can demand significant computational resources, rendering them impractical for large-scale deep learning applications. Its time complexity generally scales poorly with the size and dimensionality of the data; when the number of simplices is given by $N$, the time complexity of the standard PH computation algorithm is $O(N^3)$ (Otter et al., 2017). As a result, there has recently been a growing need for alternative features capable of capturing topological information in a computationally efficient manner. The *Euler Characteristic Curve* (ECC) is one such feature that can be computed without the need for PH calculation. Due to their ability to drastically boost computational efficiency, ECC-based descriptors have recently received increased attention (e.g., Beltramo et al., 2021; Chen et al., 2022; Dłotko & Gurnari, 2023; Hacquard & Lebovici, 2023; Malott & Wilsey, 2023; Jiang et al., 2023; Richardson & Werman, 2014; Laky & Zavala, 2024). However, these descriptors have been primarily utilized in a static manner within the framework of feature engineering. A more recent contribution in this field is the Differentiable Euler Characteristic Transformation (DECT) (Röell & Rieck, 2024), though detailed analytical explorations were not conducted. As indicated by its name, DECT utilizes ECT; a collection of ECCs computed from various directions (Turner et al., 2014).

In this paper, we aim to develop a novel computationally efficient ECC-based topological layer that facilitates integration with general deep learning models via stable backpropagation. The preceding work most closely related to ours is the recent developement of Kim et al. (2020) and Röell & Rieck (2024). Kim et al. (2020) adopted persistence landscapes to construct a differentiable topological layer. Notwithstanding its merits, persistence landscapes inherit the high computational complexity of PH, and their gradients can often be highly sparse, lacking substantial information (see Figure 5 in Appendix). The proposal for a topological layer utilizing DECT (Röell & Rieck, 2024) relies specifically on the height filtration, which works best with graphs and meshes, but not necessarily with other data structures; it may be applicable to point clouds, for example, yet with a compromise in connectivity information. Moreover, Röell & Rieck (2024) achieve differentiability of ECT by employing a sigmoid approximation, which may result in inconsistent gradients or even vanishing gradient problems when evaluated at discretized points; this will be further discussed in Section 4.

This work proposes a novel fast and robust ECC-based topological layer, `ECLayr`, designed to address all the aforementioned drawbacks comprehensively. Our proposed method obviates the need for PH calculations, thereby significantly enhancing computational efficiency while preserving the capability to extract key topological information from underlying data structures. `ECLayr` is capable of utilizing generic filtrations, exhibiting versatility across various data modalities without necessitating data preprocessing or resorting to a particular filtration. Importantly, we introduce a novel approach for stable backpropagation with respect to the layer input, addressing the inconsistent/vanishing gradient issues associated with the sigmoid approximation in DECT. We also provide a stability analysis, showing that the proposed layer is robust against noise and outliers. Our experimental analysis show that `ECLayr` delivers performance comparable to state-of-the-art PH-based topological layers, while being significantly faster by several orders of magnitude. Using our proposed backpropagation algorithm, `ECLayr` exhibits improved performance and greater efficiency than DECT. We further demonstrate its versatility through the application on topological autoencoders.

## 2 MATHEMATICAL BACKGROUND

This section provides a brief overview of the essential tools in TDA used throughout the development, as well as some notations. For further information, see, for example, Hatcher (2002); Edelsbrunner & Harer (2010); Chazal & Michel (2021); Kaczynski et al. (2004).

**Simplex and Simplicial Complex.** Let $u_0, \ldots, u_k$ be affinely independent points in $\mathbb{R}^d$. A *k-simplex* is the convex hull of the $k + 1$ points, $\sigma_k = \text{conv}\{u_0, \ldots, u_k\}$ (e.g., 0-simplex is a vertex, 1-simplex is an edge, 2-simplex is a triangle, etc.). The *dimension* of $\sigma_k$ is $k$. $\tau$ is a *face* of $\sigma_k$ if it is a convex hull constructed from any non-empty subset of the $k+1$ points of $\sigma_k$. A *simplicial complex* $K$ is a finite collection of simplices such that (i) the face of any simplex in $K$ is also in $K$, and (ii) the intersection of two simplices in $K$ is either empty or a face of both simplices. Commonly used simplicial complexes include the Vietoris-Rips complex and the Alpha complex (see Appendix A).

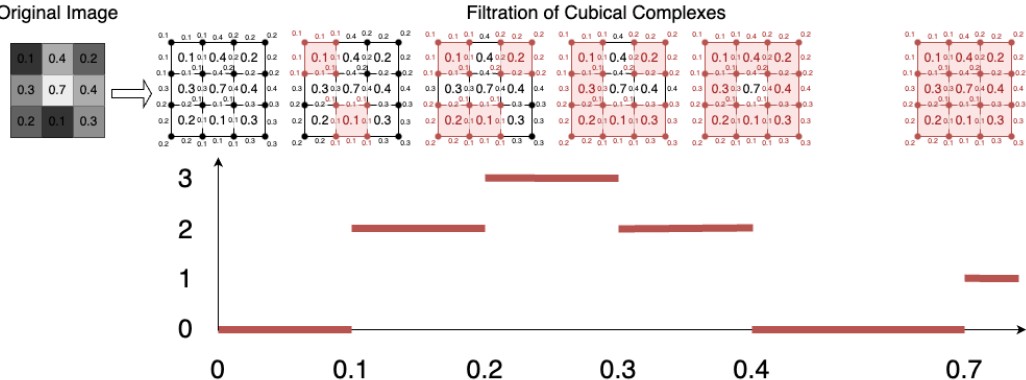

Figure 1: Computation of ECC using sublevel set filtration in filtered cubical complex.

**Filtration.** A *filtration* $\mathcal{F} = \{K(a) \subset K | a \in \mathbb{R}\}$ is a collection of nested simplicial complexes that satisfy $K(a) \subset K(b)$ whenever $a \leq b$. A typical way of constructing a filtration is to use a monotonic filtration function $f : K \rightarrow \mathbb{R}$. $f$ is monotonic in the sense that $f(\tau) \leq f(\sigma)$ whenever $\tau$ is a face of $\sigma$. By defining $K(a) := f^{-1}(-\infty, a]$, we have $K(a) \subset K(b)$ whenever $a \leq b$.

**Cubical Complex.** Cubical complex is an analogy of simplicial complex that consists of $k$-cubes (e.g., vertices, edges, squares, cubes, etc.). It provides a suitable framework for analyzing data that is naturally aligned with a grid structure (e.g., digital images). An *elementary interval* is an interval of form $I = [l, l+1]$ or $I = [l, l]$ for some $l \in \mathbb{Z}$, where the former interval is called *nondegenerate* and the latter *degenerate*. An *elementary cube* is the finite product of elementary intervals, i.e., $Q = I_1 \times I_2 \times \cdots \times I_n$. The *dimension* of $Q$ is the number of nondegenerate elementary intervals in the product. $P$ is a *face* of $Q$ if $P \subset Q$ where $P$ and $Q$ are both elementary cubes. A *cubical complex* $K$ is a finite collection of elementary cubes such that the face of any cube in $K$ is also in $K$. A *filtered* cubical complex can be constructed by assigning a filtration value to each of the cubes (see Appendix B for details).

**Euler Characteristic.** The *Euler characteristic* is a topological invariant that provides a single number summarizing the essential topological[1] features of data. Given a simplicial or cubical complex $K$, it is defined as the alternating sum of the number of $k$-simplices or $k$-cubes in $K$. It can equivalently be defined as an alternating sum of Betti numbers.

$$\chi(K) = \sum_{k=0}^{\infty} (-1)^k |K^k| = \sum_{k=0}^{\infty} (-1)^k \beta_k, \tag{1}$$

where $K^k$ is the set of $k$-dimensional simplices or cubes in $K$, $|K^k|$ is its cardinality, and $\beta_k$ is the $k$-th Betti number of $K$. We can obtain an *Euler Characteristic Curve* (ECC) $\mathcal{C} : \mathbb{R} \rightarrow \mathbb{R}$ by computing the Euler characteristic along a filtration, where the $x$-axis corresponds to the filtration values and $y$-axis corresponds to the Euler characteristic of the subcomplex at a given filtration value, i.e., for $t \in \mathbb{R}$, $\mathcal{C}(t) = \chi(K(t))$ (see Figure 1).

## 3 LAYER CONSTRUCTION

The construction of `ECLayr` involves two steps: (i) computing the ECC from input data, and (ii) passing the ECC through a differentiable map. To compute ECC, we consider an alternative representation of the Euler characteristic. Let us denote $K(t)$ as the subcomplex of $K$ at a given filtration value $t$. Then, the Euler characteristic of $K(t)$ in equation 1 can be equivalently defined as

$$\chi(K(t)) = \sum_{k=0}^{\infty} (-1)^k \sum_{\sigma \in K^k} \mathbb{1}\left[f_\sigma \leq t\right], \tag{2}$$

---

[1]Depending on the filtration, both PH and ECC can capture geometric information as well.

---

**Algorithm 1:** Computation of Euler Characteristic Curve: $X \to \mathcal{E}$

---

**1** **Hyperparameters:** $T_{min}, T_{max}, v$
**2** **Input:** $X$
**3** Choose a simplicial complex suitable for the input data and build a filtration
**4** Set $tseq = \{t_1, \ldots, t_v\}$, a sequence of $v$ evenly-spaced discretized points from $T_{min}$ to $T_{max}$
**5** Initialize $\mathcal{E} = (0, \ldots, 0) \in \mathbb{R}^v$, which are values corresponding to locations in *tseq*
**6** **for** $\sigma \in K$ **do**
**7**     **if** $f_\sigma > T_{max}$ **then**
**8**        continue
**9**     $t^* \leftarrow \min\{t_i \in tseq | t_i > f_\sigma\}$
**10**     $\mathcal{E}(t^*) \leftarrow \mathcal{E}(t^*) + (-1)^{\dim(\sigma)}$
**11** $\mathcal{E} \leftarrow \text{cumsum}(\mathcal{E})$
**12** **Output:** $\mathcal{E} \in \mathbb{R}^v$

---

where $f_\sigma$ is the filtration value of $\sigma$. The equivalence between equation 1 and equation 2 is straightforward, as the sum of indicator functions is identical to the number of $k$-simplices in the subcomplex $K(t)$. To simplify notation, let $X$, $\mathcal{E}$, and $\mathcal{O}_\theta$ represent the input, vectorized ECC, and output of our layer, respectively.

### 3.1 COMPUTATION OF ECC: $X \to \mathcal{E}$

Before calculating ECC from input data, a filtration must be defined by choosing an appropriate simplicial complex $K$ and a function $f : K \to \mathbb{R}$. This is often data-dependent; Vietoris-Rips or Alpha complexes are commonly used for point clouds while sub/superlevel set filtrations on filtered cubical complexes are a natural choice for images. Upon constructing a filtration, we proceed to obtain the vectorized approximation of ECC based on equation 2. First, we set a closed interval $[T_{min}, T_{max}]$ and sample $v$ evenly-spaced grid points ranging from $T_{min}$ to $T_{max}$. We denote these discretized points as $tseq = \{t_1, \ldots, t_v\}$, where $t_1 = T_{min}$ and $t_v = T_{max}$. Our objective is to derive a vector $\mathcal{E}$ containing the Euler Characteristics of each subcomplex $K(t_i)$; $\mathcal{E} = (\chi(K(t_1)), \ldots, \chi(K(t_v)))$. This vector serves as a finite sample approximation of the ECC function $\mathcal{C}$. To compute $\mathcal{E}$, we begin by initializing $\mathcal{E}$ as a zero vector of size $v$. Next, we iterate over all simplices $\sigma \in K$ and perform the following steps: (i) find $t^* = \min\{t_i \in tseq | t_i > f_\sigma\}$, which denotes the smallest grid point that is larger than the filtration value of $\sigma$, and (ii) add $(-1)^{\dim(\sigma)}$ to $\mathcal{E}(t^*)$. If $f_\sigma$ exceeds the upper bound $T_{max}$ and $t^*$ cannot defined, we proceed to the subsequent simplex in the iteration. Once the iteration is terminated, we return the cumulative sum of $\mathcal{E}$ for each point $t_i$. The resulting output $\mathcal{E}$ is a vector in $\mathbb{R}^v$. The procedure is summarized in Algorithm 1.

**Time Complexity.** Given a filtration, computation of ECC (Steps 6 to 12 in Algorithm 1) requires $O(N + v)$ time, where $N$ is the number of simplices and $v$ is the number of grid points. This shows a substantial improvement over the $O(N^3)$ of the standard PH computation. In our experimental analysis in Section 6.1, we empirically demonstrate that our proposed layer reduces the computational complexity up to several orders of magnitude compared to PH.

### 3.2 COMPUTATION OF LAYER OUTPUT: $\mathcal{E} \to \mathcal{O}_\theta$

By definition, ECC depends on the number of generators (Betti numbers), even if they are small (noise) generators. This implies that ECC may potentially contain some noise information. Thus, we do not use ECC directly, but employ a differentiable parametrized map $g_\theta$ to project ECC to a learnable task-optimal representation. Given $\mathcal{E} \in \mathbb{R}^v$ and an output dimension $m$, the map $g_\theta : \mathbb{R}^v \to \mathbb{R}^m$ takes $\mathcal{E}$ as input and outputs $\mathcal{O}_\theta \in \mathbb{R}^m$. There are no restrictions regarding the structure of $g_\theta$ as long as differentiability with respect to $\theta$ is guaranteed. In this paper, we use a sequence of fully connected layers and Relu nonlinearity functions to construct $g_\theta$.

# 4 STABLE BACKPROPAGATION

We first present the differentiability result for ECC. By applying chain rule, we decompose the derivative of Euler characteristic into two elements: (i) derivative of the filtration value with respect to input $X$, and (ii) derivative of the indicator function with respect to the filtration value, as shown below.

$$\frac{\partial \chi(K(t))}{\partial X} = \sum_{k=0}^{\infty}(-1)^k \sum_{\sigma \in K^k} \underbrace{\frac{\partial f_\sigma}{\partial X}}_{(i)} \underbrace{\frac{\partial \mathbb{1}(f_\sigma \leq t)}{\partial f_\sigma}}_{(ii)}.$$

The term (i) depends on the specific choice of filtration. While our framework allows the use of arbitrary differentiable filtration, here we focus on three widely-used filtrations; we provide results for Vietoris-Rips, Alpha, and sub/superlevel set filtration on filtered cubical complex in Appendix C.

## 4.1 GRADIENT INCONSISTENCY IN SIGMOID APPROXIMATION

The key barrier in achieving differentiability arises from the discontinuous nature of the indicator function $\mathbb{1}(f_\sigma \leq t)$ in the term (ii). To bypass the need for direct differentiation, Röell & Rieck (2024) adopted a smooth approximation by substituting the indicator function with a sigmoid function $S(\lambda(t - f_\sigma))$, where the extra hyperparameter $\lambda$ controls the precision of approximation. Despite being theoretically sound, such smoothing-based approximation poses a practical problem in backpropagation procedures as values are evaluated over a finite set of discretized grid points rather than a continuous domain. When a filtration value $f_\sigma$ lies between grid points, the gradient with respect to $f_\sigma$ is not evaluated at the precise location, but at its neighboring grid points. Thus, the magnitude of gradient varies depending on the proximity of $f_\sigma$ to its adjacent grid points. This will be referred to as *gradient inconsistency* (see Figure 2).

Especially, we show that the sigmoid approximation is prone to gradient vanishing problems when insufficient $v$ leads to excessive spacing between grid points, or when $\lambda$ is too large (see Figure 2-(c)). To formally state this issue, we suppose the gradients of the indicator function $\mathbb{1}(f_\sigma \leq t)$ with respect to $f_\sigma$ are approximated on a fixed grid $tseq = \{t_1, \ldots, t_v\}$ with $\Delta t := \frac{t_{i+1} - t_i}{2}$ being equal. Let $S'^{tseq}_{\lambda, f_\sigma} \in \mathbb{R}^v$ be the gradient vector of the sigmoid function $S(\lambda(t - f_\sigma))$ computed at $t_1, \ldots, t_v$, i.e.,

$$S'^{tseq}_{\lambda, f_\sigma} = \frac{\partial S(\lambda(t - f_\sigma))}{\partial f_\sigma}\big|_{t=t_1,\ldots,t_v}.$$

The next proposition shows that the local gradient of the sigmoid approximation can approach arbitrarily close to zero, regardless of its true value.

**Proposition 4.1.** *When $\frac{\partial S(\lambda(t-f_\sigma))}{\partial f_\sigma}$ is viewed as a function of $t$, then its $L_\infty$ norm is computed as*

$$\left\|\frac{\partial S(\lambda(t - f_\sigma))}{\partial f_\sigma}\right\|_\infty = \frac{\lambda}{4},$$

*while its discretization over $tseq$ is $L_\infty$ bounded as*

$$\left\|S'^{tseq}_{\lambda, f_\sigma}\right\|_\infty \leq \lambda S(\lambda d(f_\sigma, tseq))\left[1 - S(\lambda d(f_\sigma, tseq))\right].$$

*So in particular when $\lambda \exp(-\lambda \Delta t) \to 0$,*

$$\inf_{f_\sigma \in [t_1 - \Delta t, t_v + \Delta t)}\left\|S'^{tseq}_{\lambda, f_\sigma}\right\|_\infty \to 0.$$

The issue of diminishing gradients during the training is particularly vexing in deep learning. As the downstream gradient is computed via a series of multiplications involving local gradients, the sigmoid approximation in ECC will eventually impede effective backpropagation.

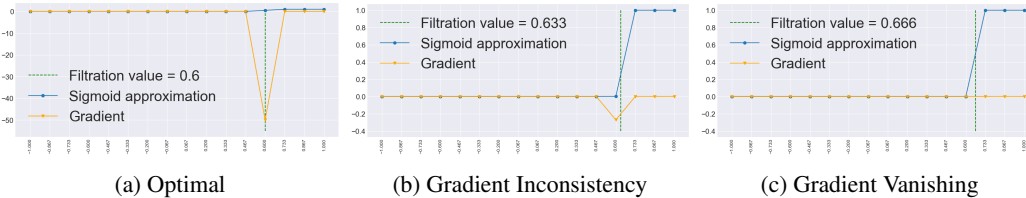

(a) Optimal       (b) Gradient Inconsistency       (c) Gradient Vanishing

Figure 2: An illustration of gradient inconsistency issues when using the sigmoid approximation with $\lambda = 200$. The values are assessed at 16 evenly-spaced points over $[-1, 1]$. In (a), the sigmoid approximation performs well when the filtration value precisely aligns with one of the grid points. In (b), however, a slight shift in the filtration value results in a significant change in the gradient. A further slight adjustment eventually leads to gradient vanishing, as shown in (c). This indicates that the gradients vary significantly when the sigmoid approximation is used for the ECC layer, depending on the position of the filtration value relative to the grid points.

### 4.2 STABLE BACKPROPAGATION VIA DISTRIBUTIONAL DERIVATIVES

To resolve the gradient inconsistency issue, here we propose an alternative approach in which we approximate the *gradient* rather than the indicator function itself. In order to compute the gradient, we resort to distributional derivatives; $\frac{\partial \mathbb{1}[f_\sigma \leq t]}{\partial f_\sigma} = -\delta(t - f_\sigma)$ where $\delta(x) = \lim_{\beta \to 0} \frac{1}{|\beta|\sqrt{\pi}} e^{-(x/\beta)^2}$ is the dirac delta, a function that has a single impulse at $x = 0$ and zero elsewhere. Since the height of this single impulse is infinite, we proceed with approximation $\max_x \frac{1}{|\beta|\sqrt{\pi}} e^{-(x/\beta)^2} = \frac{1}{|\beta|\sqrt{\pi}}$ where $\beta$ is a hyperparameter that determines the height of the spike. Namely, the estimated gradient will always be $\frac{1}{|\beta|\sqrt{\pi}}$ at the impulse point and zero elsewhere. It is also critical to ensure that the gradient does not leak; while the approximated gradient has a single impulse at $t = f_\sigma$, it may not necessarily correspond to the predefined positions $tseq$. Thus, we shift the location of impulse so that it aligns with one of the points in $tseq$. Recall from Section 3.1 that for a given simplex $\sigma$, $t^* = \min\{t_i \in tseq | t_i > f_\sigma\}$ is the grid point where the jump is reflected during the forward pass. Consequently, we backpropagate the gradient to the identical location $t^*$. In this formulation, the gradient invariably traverses one of the discretized locations, unless it was ignored during the forward pass. As a result, we can assure that a consistent gradient value is properly backpropagated to the preceding layer. The following proposition shows that our proposed method prevents the gradient vanishing issues associated with the sigmoid approximation.

**Proposition 4.2.** *Let* $\hat{\delta}_{\beta,f_\sigma}^{tseq} \in \mathbb{R}^v$ *be our gradient approximation of* $\frac{\partial \mathbb{1}(f_\sigma \leq t)}{\partial f_\sigma}$ *computed at* $t_1, \ldots, t_v$, *so* $\hat{\delta}_{\beta,f_\sigma}^{tseq}$*'s jth element is* $-\frac{1}{\beta\sqrt{2\pi}}$ *if* $f_\sigma \in [t_{j-1}, t_j)$, *and other elements are* 0. *Then the* $L_\infty$ *norm of* $\hat{\delta}_{\beta,f_\sigma}^{tseq}$ *is given as*

$$\left\| \hat{\delta}_{\beta,f_\sigma}^{tseq} \right\|_\infty = \frac{1}{\beta\sqrt{2\pi}}.$$

Aside from the issue of diminishing gradients, we have shown that our proposed approaches can achieve much lower errors in approximating true gradient values. Specifically, with $\Delta t :=$ $\frac{t_{i+1} - t_i}{2}, \forall i$, we show that by letting $\beta = \frac{\sqrt{\pi}}{2\Delta t}$, our proposed methods may attain consistency. We refer to Appendix D for detailed theoretical results.

**Time Complexity of Sigmoid Approximation.** Computation of ECC via sigmoid approximation requires $O(vN)$ time, as the sigmoid function must be applied to every $t_i \in tseq$ during each iteration across all simplices $\sigma \in K$. Our backpropagation method enables the use of Algorithm 1 during forward pass, achieving enhanced efficiency of $O(N + v)$.

## 5 STABILITY THEOREM

An essential benefit of using a topological layer is its robustness against noise. Extending the results of Dłotko & Gurnari (2023), we can establish a stability property for the layer output with respect to changes in the input. For notation, let $X, X'$ be two distinct inputs, and $f_X, f_{X'}$ be corresponding

filtration functions on fixed simplicial complexes $K$, $K'$, respectively. Let $\mathcal{D}_k(X), \mathcal{D}_k(X')$ be corresponding $k$-dimensional persistence diagrams, and let $\mathcal{C}_X, \mathcal{C}_{X'} : \mathbb{R} \to \mathbb{R}$ be corresponding ECC functions. See Appendix A for the definition of persistence diagrams and Wasserstein distance.

We first see the relation between the final layer output and ECC functions.

**Proposition 5.1.** *Let $t_1^* < t_2^* < \cdots < t_w^*$ be unique values of all births and deaths in $\{\mathcal{D}_k(X), \mathcal{D}_k(X') : k \geq 0\}$, and let $tseq = (t_1, \ldots, t_v)$. Suppose there exists $\Delta t > 0$ satisfying that $\Delta t < t_{j+1} - t_j$ and $\Delta t < t_{j+1}^* - t_j^*$. Let $g_\theta$ be L-Lipschitz with respect to $\|\cdot\|_1$-norm, i.e., $\|g_\theta(x) - g_\theta(y)\|_1 \leq L \|x - y\|_1$. Then*

$$\|\mathcal{O}_\theta(X) - \mathcal{O}_\theta(X')\|_1 \leq \frac{2L}{\Delta t} \|\mathcal{C}_X - \mathcal{C}_{X'}\|_1.$$

Hence what we really need to establish is the stability of ECC functions. We first address the most general stability result with respect to the 1-Wasserstein distance of the persistence diagrams of input, which is directly from Dłotko & Gurnari (2023).

**Proposition 5.2** (Dłotko & Gurnari (2023), Proposition 3.2)**.**

$$\|\mathcal{C}_X - \mathcal{C}_{X'}\|_1 \leq 2 \sum_{k=0}^{\infty} W_1(\mathcal{D}_k(X), \mathcal{D}_k(X')).$$

The behavior of the 1-Wasserstein distance $W_1(\mathcal{D}_k(X), \mathcal{D}_k(X'))$ is in general complicated and difficult to analyze. It is possible to further upper bound this by the difference of the filtration functions $f_X$ and $f_{X'}$. The difference is represented as $L_\infty$ distance below, but there is a more general version of Theorem 5.3 as well.

**Theorem 5.3.** *Suppose $K = K'$ and is a finite simplicial complex or cubical complex. Then there exists a constant $C_K$ only depending on $K$ such that*

$$\|\mathcal{C}_X - \mathcal{C}_{X'}\|_1 \leq C_K \|f_X - f_{X'}\|_\infty.$$

Theorem 5.3 provides a stability result whose relation to the difference of the input is clear, and also applicable to general filtration functions. Since we use DTM functions in Section 6, we present a specific result for DTM.

**Corollary 5.4.** *Suppose $K$ is a finite cubical complex, and $f_X$, $f_{X'}$ are restrictions of DTM functions $d_{P_X, m_0}, d_{P_{X'}, m_0}$ to $K$, where $P_X, P_{X'}$ are empirical distributions on $X$ and $X'$, respectively. (for detailed meaning, see Appendix G.) Then*

$$\|\mathcal{C}_X - \mathcal{C}_{X'}\|_1 \leq \frac{C_K}{\sqrt{m_0}} W_2(P_X, P_{X'}).$$

Due to the inherent reliance of Euler characteristics on even small generators, we note that the above stability results in terms of the Wasserstein distance are less strict than those bounded by the Bottleneck distance in Kim et al. (2020). Thus, Euler characteristic-based descriptors compromise stability in order to attain computational efficiency over PH-based descriptors.

# 6 EXPERIMENTS

To showcase the versatility and effectiveness of our layer, we conduct a series of experiments. First, we demonstrate the computational efficiency of our approach by measuring runtime metrics across different datasets. Next, we proceed to implement a topological autoencoder using point clouds to illustrate an application of our layer in imposing topological constraints on the latent space. Finally, we perform classification tasks on two image datasets: MNIST and Br35H. The first image classification task shows that our layer can effectively mitigate information loss under conditions of data scarcity or data contamination. The subsequent experiment highlights the distinct advantages of our layer by performing operations on moderately high-dimensional data, which would otherwise necessitate intensive computation for PH, rendering it impractical for real-world applications. All experiments are implemented using GUDHI (The GUDHI Project, 2021) and Pytorch. Here, we present only a partial summary of the experimental findings; for comprehensive results and detailed descriptions of the architecture and hyperparameter selection, please refer to Appendix I.

| Model | Data (Number of samples) | | |
|-------|---------------|-------------|--------------|
|       | MNIST (60000) | Br35H (209) | Synth. (1000) |
| ECC   | 3.129 sec     | 0.458 sec   | 2.17 sec     |
| PH    | 33.700 sec    | 11.033 sec  | 59.288 sec   |

Table 1: Average runtime performance per iteration (in seconds).

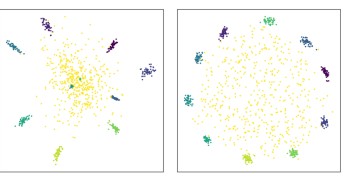

Autoencoder        Ours

Figure 3: Latent representations of Spheres data.

### 6.1 COMPUTATIONAL EFFICIENCY

In this section, we analyze the empirical time complexity of our method in comparison to PH. The time complexity of each topological descriptor is assessed by measuring the runtime for a complete iteration through the training dataset, averaged over 10 repetitions. PH computes the persistence diagram using the GUDHI package, while ECC computes the vectorized approximation of ECC using Algorithm 1. In order to investigate how each descriptor scales with increasing data dimensions, we additionally generate a synthetic dataset containing 1000 samples of size $224 \times 224$, where each pixel is randomly sampled from a uniform distribution. The experiment results for different datasets are provided in Table 1. We can observe that PH scales poorly as the dimension of the data increases. Considering the additional computation often required to transform persistence diagrams into alternative representations better suited for machine learning, our approach offers a significant benefit over all PH-based metrics in terms of computation, both in theory and in practice.

### 6.2 TOPOLOGICAL AUTOENCODER

The idea of imposing topological constraints on the latent space was first explored by Hofer et al. (2019); Moor et al. (2020). Whereas existing works rely on a topology-based loss term to regularize the latent space, our formulation allows for the utilization of standard loss functions, such as Mean Squared Error (MSE) or Mean Absolute Error (MAE) to achieve a similar goal. Inspired by the stability results regarding $L_1$ distance in Section 5, we employ the MAE loss between ECC of input and ECC of latent representation as our topological constraint. The respective ECCs are computed using Vietoris-Rips filtration, with maximum dimension set to 1. For the experiment, we use the synthetic Spheres dataset from Moor et al. (2020). The dataset consists of ten 100-spheres with radius $r = 5$ enclosed by one larger 100-sphere with radius $= 25$, all embedded in 101-dimension. The ten smaller spheres are shifted in random directions according to Gaussian noise.

**Result.** We discover that our approach effectively preserves the underlying shape of the encompassing sphere (yellow points in Figure 3), in contrast to the vanilla autoencoder, which loses this shape. Moreover, it constrains the smaller spheres to remain on the boundary of the encompassing sphere, whereas in the vanilla autoencoder, numerous smaller circles lie far beyond the boundaries of the encompassing circle. However, with this simplistic architecture, its capacity to comprehensively articulate the nested relationship inherent in the data was somewhat restricted. While our method demonstrates capability of regularizing the latent space, we do not claim superiority over alternative approaches. Rather, we present it as a motivating example of how topological characterization in the latent space can be promoted via simple standard loss functions.

### 6.3 CLASSIFICATION AGAINST DATA SCARCITY AND DATA CONTAMINATION

Our primary interest in this section is to demonstrate that our layer can effectively mitigate information loss under conditions of data scarcity and data contamination. For such purpose, we consider two scenarios on the MNIST dataset. In the first scenario, we restrict the training data to $100, 300, 500, 700$ and $1000$ samples to observe how model performance changes with data size. In the second scenario, we consider a corruption and noise process where the pixels are randomly omitted and subsequently contaminated by random noise between 0 and 1 with probability $0.05, 0.1, 0.15$, and $0.2$.

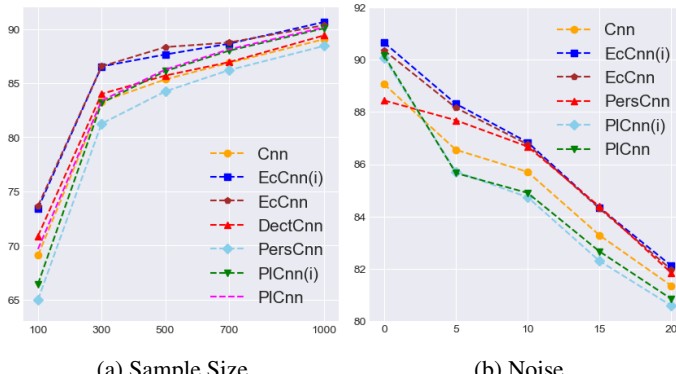

|          | (a) Sample Size | (b) Noise |
|----------|:---------------:|:---------:|

Figure 4: MNIST test accuracy: (a) Performance across different sample sizes; (b) Performance in the presence of noise.

| Model | Runtime |
|-------|---------|
| CNN | 0.088 sec |
| CNN + EC(i) | 0.340 sec |
| CNN + EC | 0.731 sec |
| CNN + DECT | 10.763 sec |
| CNN + Pers | 11.170 sec |
| CNN + PL(i) | 7.046 sec |
| CNN + PL | 14.03 sec |

Table 2: Average runtime per epoch over 1000 MNIST data without noise (in seconds).

**Experimental Setup.** To impartially illustrate the advantages of our layer, we purposefully retain a simple experimental setting. The base model consists of two CNN layers followed by two fully connected layers. We compare the performance of our proposed layer with a base model, and two other topological layers applicable to image datasets: PersLay (Carrière et al., 2020) and PLLay (Kim et al., 2020). For the data scarcity scheme, we additionally implement an ECLayr using the sigmoid approximation (denoted as CNN + DECT) previously applied by (Röell & Rieck, 2024). For all topological layers, we place a parallel layer at the beginning of the network (referred to as CNN + EC(i), Pers, PL(i)). For topological layers that allow backpropagation, we add an additional layer after the last convolutional layer (referred to as CNN + EC, DECT, PL). We implement superlevel cubical filtration for the experiment with varying data size. In the experiment involving different noise levels, we employ the DTM filtration, a tool used in TDA to robustly extract topological features in the presence noise (see Appendix A for further details). As DTM can control the level of locality when extracting topological information, we place two parallel topological layers with different scales at the beginning of the network when using DTM filtration. Utilizing a very simple model on limited training samples, we observed random failures across all models with outliers significantly affecting the outcome. To remove the influence of outliers and solely evaluate model performance, we repeat each experiment 15 times and select the top 10 test accuracies for assessment. 30% of the training data is used as a validation set, while model performance is evaluated on the full test set.

**Result.** In Figure 4, we observe that by utilizing topological information, the performance of `ECLayr` consistently surpasses the baseline in all scenarios. Surprisingly, we notice that `ECLayr` outperforms PH-based models despite the fact that PH is more informative than Euler Characteristics. We speculate that this phenomenon stems from an optimization process, coupled with the consideration that solely macroscopic topological features are adequate for this uncomplicated dataset. PH provides multiple summaries for each homology dimension, which complicates optimization in scenarios with limited data, whereas ECC yields a single summary for all dimensions. Consequently, the simplicity of ECC renders our layer more appropriate for scenarios with insufficient data. We also observe that our model outperforms ECLayr with sigmoid approximation, supporting the use of our proposed stable backpropagation method. Furthermore, `ECLayr` exhibits resistance to approximately $5 \sim 10\%$ of data contamination compared to the baseline model. Nevertheless, the inherent dependence of ECC on even small generators result results in our layer exhibiting reduced noise resistance compared to, for instance, PersLay. Runtime metrics are provided in Table 2. Our method scales approximately 20 to 30 times faster compared to PH-based methods, highlighting the significant improvement in computational efficiency. The high runtime of CNN + DECT results from the increased time complexity of $O(vN)$ when using sigmoid approximation.

### 6.4 CLASSIFICATION ON MODERATELY HIGH-DIMENSIONAL DATA

A significant drawback of PH is that its time complexity generally scales poorly with the dimension of data, rendering it impractical for high-dimensional data applications. Conversely, the computa-

| Model | Test Accuracy | Runtime |
|---|---|---|
| ResNet | 81.705 ($\pm 2.899$) | 0.110 sec |
| ResNet + EC(i) | 82.936 ($\pm 2.520$) | 0.415 sec |
| ResNet + EC | **84.351** ($\pm 3.308$) | 0.470 sec |

Table 3: Br35H test accuracy and runtime per epoch.

tional efficiency of ECC enables the use of moderately high-dimensional data without compromising significant computational costs. We demonstrate that our layer can enhance model performance by effectively exploiting topological information using a real-world dataset with dimension that would normally require intense computation for PH applications.

**Experimental Setup.** We conduct a binary classification task of detecting brain tumors on the Br35H dataset. The Br35H dataset consists of 3000 brain MRI images that have different size for each dimension and a varying number of channels. We preprocess the data by cropping along the shorter dimension, resizing it to $112 \times 112$, and converting it to grayscale. ResNet18 (He et al., 2016) is employed as a baseline model, with an additional fully connected layer of size 64 appended at the end of the network. For ResNet + EC(i), we add a parallel `ECLayr` at the beginning of the network and concatenate the output with the residual layer output before feeding to the fully connected layer. For ResNet + EC, we place an additional `ECLayr` before the first residual layer. As the task is a simple binary classification problem, we only use $10\%$ of the data as training samples and $30\%$ of training data is used for validation. Our training scheme utilizing limited data mirrors real-world challenges, as access to medical data is often limited and costly. Each simulation is repeated 10 times, with the average test accuracy and average runtime per epoch reported in Table 3.

**Result.** The results in Table 3 show that our layer can enhance model performance while maintaining manageable computational costs on moderately high dimensional data. Furthermore, it suggests that our layer can be effectively integrated with large models such as ResNet for practical usage. Another interesting observation is that using an additional `ECLayr` before the first residual layer yields further improvement in performance. Fully exploiting the computational efficiency of ECC, our layer facilitates operations on moderately high-dimensional data that would be impractical for PH, highlighting the significance of `ECLayr` for real-world applications.

## 7 DISCUSSION

`ECLayr` is a novel topological layer that offers computationally efficiency and stable backprop-agation, allowing for seamless integration into a wide range of deep learning architectures while enhancing both robustness and convergence behavior. Our proposed layer can be used generically for an extensive variety of data structures as long as the filtration is differentiable with respect to the input data. Nonetheless, there are some important caveats and limitations which should be addressed. First, while ECCs offer computational efficiency, PH-based summaries provide more detailed, multi-scale topological information. Understanding this tradeoff is essential. Therefore, our proposed `ECLayr` is particularly well-suited for applications where computational efficiency is prioritized over detailed topological insights. Moreover, as discussed in greater detail in Section 5, ECCs are topologically weaker invariants compared to PHs. Consequently, the ECC-based layer generally exhibits less robustness than the PH-based layers. Next, as with other topological layers, further research is necessary to achieve successful systematic hyperparameter exploration. Finally, extending our analysis to other filtrations, such as the clique complex of a multigraph, and applying `ECLayr` to time-series embeddings (Kim et al., 2018; Umeda, 2017) would be a valuable direction for future research, which could further demonstrate the versatility of our proposed methods.

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

# APPENDIX

## A MORE BACKGROUNDS IN TOPOLOGICAL DATA ANALYSIS

We briefly review basic concepts in Topological Data Analysis that are needed to develop stability results in Section 5 of this paper, mainly coming from Kim et al. (2020). We refer interested readers to Chazal & Michel (2021); Hatcher (2002); Edelsbrunner & Harer (2010); Chazal et al. (2009; 2016a) for details and formal definitions.

**Vietoris-Rips Complex.** Let $X$ be a finite set of points in $\mathbb{R}^d$. For $r > 0$, the *Vietoris-Rips complex* is a collection of simplices where the distance between any two vertices is smaller than $2r$:

$$\text{Rips}(r) = \{\sigma \subset X | d(u_i, u_j) < 2r, \forall u_i, u_j \in \sigma\}.$$

Notice that $\text{Rips}(r_1) \subset \text{Rips}(r_2)$ when $r_1 \leq r_2$. Thus, we can build a filtration on the Vietoris-Rips complex by monotonically increasing $r$.

**Alpha Complex.** Let $X$ be a finite set of points in $\mathbb{R}^d$. For each $u_i \in X$, the *Voronoi cell* of $u_i$ is the set of points that are closest to $u_i$; $V_{u_i} = \{x \in \mathbb{R}^d | d(u_i, x) \leq d(u_j, x), \forall u_j \in X, u_j \neq u_i\}$. For $r > 0$ and each $u_i \in X$, let us denote the closed $r$-*ball* with center $u_i$ and radius $r$ as $B_{u_i}(r)$. Then, we define $R_{u_i}(r) = B_{u_i}(r) \cap V_{u_i}$, which is the intersection of each $r$-ball with its corresponding Voronoi cell. The *Alpha complex* is a collection of simplices such that all $R_{u_i}(r)$ of the vertices in the simplex have an intersection:

$$\text{Alpha}(r) = \{\sigma \subset X | \cap_{u_i \in \sigma} R_{u_i}(r) \neq \emptyset\}.$$

Similar to the Vietoris-Rips complex, we can build a filtration on the Alpha complex by monotonically increasing $r$.

**Persistent Homology and Persistence Diagram.** *Persistent homology* is a multiscale approach to represent the topological features of the complex $K$, and can be represented in the persistence diagram. For a filtration $\mathcal{F}$ and for each nonnegative $k$, we keep track of when $k$-dimensional homological features (e.g., 0-dimension: connected component, 1-dimension: loop, 2-dimension: cavity,...) appear and disappear in the filtration. If a homological feature $\alpha_i$ appears at $b_i$ and disappears at $d_i$, then we say $\alpha_i$ is born at $b_i$ and dies at $d_i$. By considering these pairs $(b_i, d_i)$ as points in the plane, one obtains the *persistence diagram* defined as follows.

**Definition A.1.** *Let* $\mathbb{R}^2_* := \{(b, d) \in (\mathbb{R} \cup \infty)^2 : d > b\}$. *A persistence diagram* $\mathcal{D}$ *is a finite multiset of* $\{(b_i, d_i) : (b_i, d_i) \in \mathbb{R}^2_*\}$.

**Wasserstein Distance.** We suggest two versions of Wasserstein distances, one is for persistence diagrams and the other is for probability measures.

We first start with Wasserstein distance for persistence diagrams. A *matching* between two persistence diagrams $\mathcal{D}_1$ and $\mathcal{D}_2$, is a subset $m \subset \mathcal{D}_1 \times \mathcal{D}_2$ such that every off-diagonal point in $\mathcal{D}_1$ and $\mathcal{D}_2$ only appears once in $m$. The *p-Wasserstein distance* between persistence diagrams is defined by

$$W_p(\mathcal{D}_1, \mathcal{D}_2) = \inf_{\text{matching } m} \left( \sum_{(x,y) \in m} \|x - y\|_\infty^p \right)^{1/p}$$

Now we see Wasserstein distance for persistence diagrams. Let $P$ and $Q$ be probability measures on $\mathcal{X}$, and let $\mathcal{J}(P, Q)$ denote all joint distributions $J$ for $\mathcal{X} \times \mathcal{X}$ that have marginals $P$ and $Q$. In other words, $(\Pi_1)_{\#} J = P$ and $(\Pi_2)_{\#} J = Q$ where $\Pi_1(x, y) = x$ and $\Pi_2(x, y) = y$, and $T_{\#} P$ is a push-forward measure of $P$, i.e., $T_{\#} P(A) = P\left(\{x : T(x) \in A\} = P(T^{-1}(A))\right)$. For $p \geq 1$, the Kantorovich, or Wasserstein, distance is

$$W_p(P, Q) = \left( \inf_{J \in \mathcal{J}(P,Q)} \int_{\mathcal{X} \times \mathcal{X}} ||x - y||^p dJ(x, y) \right)^{1/p}.$$

**Gromov-Hausdorff distance.** The *Hausdorff distance* is on sets embedded in the same metric spaces. This distance measures how two sets are close to each other in the embedded metric space. When $S \subset \mathbb{X}$, we denote by $S^r$ the $r$-neighborhood of a set $S$ in $\mathbb{R}^d$, i.e. $S^r = \bigcup_{x \in S} B_x(r)$.

**Definition A.2** (Hausdorff distance (Burago et al., 2001, Definition 7.3.1)). *Let $X, Y \subset \mathbb{X}$ be subsets of $\mathbb{R}^d$. The* Hausdorff distance *between $X$ and $Y$, denoted by $d_H(X, Y)$, is defined as*

$$d_H(X, Y) := \inf \left\{ r > 0 : \ X \subset Y^r \ and \ Y \subset X^r \right\}.$$

The notion of the Hausdorff distance can be generalized to the comparison of any pair of metric spaces. The *Gromov-Hausdorff distance* measures how two sets are far from being isometric to each other.

**Definition A.3** ((Burago et al., 2001, Definition 7.3.10)). *Let $X$ and $Y$ be two metric spaces. The* Gromov-Hausdorff distance *between $X$ and $Y$, denoted by $d_{GH}(X, Y)$, is defined as*

$$d_{GH}(X, Y) := \inf\{d_H(X', Y') : there \ exists \ a \ metric \ space \ Z \ and \ X', Y' \subset Z$$
$$with \ X, Y \ isometric \ to \ X', Y', \ respectively.\}$$

**Distance to measure.** Distance to measure (DTM) (Chazal et al., 2011; 2016b; Anai et al., 2020) is a distance-like function[2] that is robust to outliers. For a probability measure $\mu$ and parameters $m_0 \in [0, 1)$ and $r \geq 1$ (default is $r = 2$), the DTM function $d_{\mu, m_0} : \mathbb{R}^d \to \mathbb{R}$ is defined as

$$d_{\mu, m_0}(x) = \left( \frac{1}{m_0} \int_0^{m_0} \delta_{\mu, m}^r(x) dm \right)^{1/r},$$

where $\delta_{\mu, m}(x) = \inf\{t > 0 | \mu(B_x(t)) > m\}$ and $B_x(t)$ is a closed $t$-ball centered at $x$. In practice, an empirical DTM is used. If input data X is considered as weights corresponding to fixed points Y,

$$\hat{d}_{m_0}(x) = \left( \frac{\sum_{Y_i \in N_k(x)} X_i' \| Y_i - x \|^r}{m_0 \sum_{i=1}^n X_i} \right)^{1/r}, \tag{3}$$

where $N_k(x)$ is a subset of $Y$ containing the $k$ nearest neighbors of $x$. $k$ is such that satisfies $\sum_{Y_i \in N_{k-1}(x)} X_i < m_0 \sum_{i=1}^n X_i \leq \sum_{Y_i \in N_k(x)} X_i$, and $X_i' = \sum_{Y_j \in N_k(x)} X_j - m_0 \sum_{j=1}^n X_j$ if at least one of $Y_i$'s is in $N_k(x)$ and $X_i' = X_i$ otherwise (see Figure 5 (b)).

When input data is considered as empirical data points, the empirical DTM becomes

$$\hat{d}_{m_0}(x) = \left( \frac{\sum_{X_i \in N_k(x)} w_i' \| X_i - x \|^r}{m_0 \sum_{i=1}^n w_i} \right)^{1/r}$$

where $N_k(x)$ is a subset of $X$ containing the $k$ nearest neighbors of $x$. $k$ is such that satisfies $\sum_{X_i \in N_{k-1}(x)} w_i < m_0 \sum_{i=1}^n w_i \leq \sum_{X_i \in N_k(x)} w_i$, and $w_i' = \sum_{X_j \in N_k(x)} w_j - m_0 \sum_{j=1}^n w_j$ if at least one of $X_i$'s is in $N_k(x)$ and $w_i' = w_i$ otherwise.

The parameter $m_0$ determines how much local/global structures should be extracted, with smaller $m_0$ corresponding to more local structures. The DTM function is differentiable (Kim et al., 2020), and adopting a sublevel or superlevel set filtration on the DTM transformed data yields a DTM filtration that is robust to outliers.

# B  CONSTRUCTING FILTERED CUBICAL COMPLEXES FROM IMAGE DATA

Let $X \in \mathbb{R}^{H \times W}$ be a 2D image. There are two methods of constructing a filtered cubical complex: T-construction and V-construction.

**T-construction** In *T-construction*, each pixel in the image is mapped to a top-dimensional cell in the cubical complex, which is a square in case of 2D images. The filtration value of each square is assigned as the intensity of its corresponding pixel, and these filtration values are recursively extended to lower dimensional cubes. The filtration value of each edge is assigned as the minimum of the filtration values of its neighboring squares. Similarly, the filtration value of each vertex is assigned as the minimum of the filtration values its neighboring edges.

---

[2]This distance function is not the distance function giving a metric between two input points such as $l_p$ distance, but rather measures a distance between a single input point and the support set of a probability distribution.

**V-construction** In *V-construction*, each pixel in the image is mapped to a vertex in the cubical complex. The filtration value of each vertex is assigned as the intensity of its corresponding pixel, and these filtration values are recursively extended to higher dimensional cubes. The filtration value of each edge is assigned as the maximum of the filtration values of its neighboring vertices. Similarly, the filtration value of each square is assigned as the maximum of the filtration values of its neighboring edges.

For both constructions, a sublevel set at a given filtration value $t$ defines a subcomplex $K(t) := \{\sigma \in K | f(\sigma) \leq t\}$; the collection of cubes with filtration value less than or equal to $t$. Consequently, a sublevel set filtration can be built by monotonically increasing $t$. A superlevel set filtration can also be obtained by applying the sublevel set filtration to a cubical complex constructed from $-X$ rather than $X$. In case of 2D images with multiple channels, such as color images represented by RGB channels where $X \in \mathbb{R}^{3 \times H \times W}$, cubical complexes are constructed independently for each channel.

## C    DERIVATIVE OF FILTRATION VALUE WITH RESPECT TO INPUT $X$: $\frac{\partial f(\sigma)}{\partial X}$

### C.1    VIETORIS-RIPS FILTRATION

Assume Vietoris-Rips general position for a point cloud $X$: (i) all points in $X$ are unique, and (ii) the length of all attaching edges are unique. The filtration value of a simplex $\sigma$ in the Vietoris-Rips filtration is half the length of the longest edge in $\sigma$. This edge is the *attaching edge* of $\sigma$, denoted as $\tau_\sigma$. Letting $x_i$ and $x_j$ be the vertices of $\tau_\sigma$, the derivatives of filtration value $f(\sigma) = \frac{\|x_i - x_j\|}{2}$ with respect to the points $x_i$ and $x_j$ are given by (Gameiro et al., 2016):

$$\frac{\partial f(\sigma)}{\partial x_i} = \frac{1}{2} \frac{x_i - x_j}{\|x_i - x_j\|}, \quad \frac{\partial f(\sigma)}{\partial x_j} = \frac{1}{2} \frac{x_j - x_i}{\|x_i - x_j\|}. \tag{4}$$

The derivatives with respect to points other than $x_i$ and $x_j$ are all zero.

### C.2    ALPHA FILTRATION

Assume Alpha general position of a point cloud $X$: (i) general position in the sense of Edelsbrunner & Mücke (1994), and (ii) filtration values of all attaching simplices are unique. In Alpha filtration, all simplices are either an attaching simplex, or a simplex attached by another simplex of higher dimension. In the latter case, filtration value of the attached simplex is given by the filtration value of its attaching simplex. The filtration value of an attaching simplex $\sigma$ is the radius of the smallest circumcircle of $\sigma$ (Edelsbrunner & Mücke, 1994; Gameiro et al., 2016) and it can be differentiated with respect to the coordinates of each of the vertices.

### C.3    SUB/SUPERLEVEL SET FILTRATION ON FILTERED CUBICAL COMPLEXES

Let us treat a 2D image $X \in \mathbb{R}^{H \times W}$ as a vector $x = (x_1, \ldots, x_{HW}) \in \mathbb{R}^{HW}$, where the elements of the vector are arranged in row-major order. Then, the derivative of the filtration value with respect to the input data can be written as

$$\frac{\partial f(\sigma)}{\partial x} = \left( \frac{\partial f(\sigma)}{\partial x_1}, \ldots, \frac{\partial f(\sigma)}{\partial x_{HW}} \right)$$

Given that the filtration value varies depending on the construction used, we provide differentiability results for both T-construction and V-constructions. For simplicity of notation, we denote $\mathcal{I} = \{1, 2, \ldots, HW\}$ as the index set.

**T-construction.** In T-construction, each pixel is mapped to a square, with the pixel intensity serving as the filtration value of the corresponding square. Thus, we first explore the scenario where $\sigma$ is a square, and then extend our analysis to lower dimensional cubes.

(i) Assume $\sigma$ is a square, i.e., $\dim(\sigma) = 2$. Let $j \in \mathcal{I}$ denote the index of the pixel in $x$ that corresponds to $\sigma$. Then, $f(\sigma) = x_j$ and thus,

$$\frac{\partial f(\sigma)}{\partial x_i} = \begin{cases} 1, & \text{if } i = j \\ 0, & \text{otherwise} \end{cases}$$

for all $i \in \mathcal{I}$.

(ii) Assume $\sigma$ is an edge, i.e., $\dim(\sigma) = 1$. Recall that $f(\sigma)$ is assigned as the minimum filtration value of its neighboring squares, which in turn is equivalent to the minimum pixel intensity of the pixels corresponding to those neighboring squares. Thus, we can identify the pixel associated with $\sigma$ by

    1. find neighboring squares of $\sigma$

    2. determine the neighboring square with minimum filtration value

    3. identify the pixel that corresponds to the square found in (2)

In step 2, multiple neighboring squares may have the same minimum filtration value. In this case, we identify the set of pixels that corresponds to all such squares. Letting $J \subset \mathcal{I}$ denote an index set labeling the members of such set of pixels,

$$\frac{\partial f(\sigma)}{\partial x_i} = \begin{cases} 1/|J|, & \text{if } i \in J \\ 0, & \text{otherwise} \end{cases}$$

for all $i \in \mathcal{I}$. Observe that when multiple pixels contribute to $\sigma$, we distribute the gradient evenly between those pixels.

(iii) Assume $\sigma$ is a vertex, i.e., $\dim(\sigma) = 0$. Recall that $f(\sigma)$ is assigned as the minimum filtration value of its neighboring edges. Therefore, once we find the neighboring edge(s) with minimum filtration value, we can repeat the process in (ii) to identify the set of pixels associated with $\sigma$. Letting $J \subset \mathcal{I}$ denote an index set labeling the members of such set of pixels,

$$\frac{\partial f(\sigma)}{\partial x_i} = \begin{cases} 1/|J|, & \text{if } i \in J \\ 0, & \text{otherwise} \end{cases}$$

for all $i \in \mathcal{I}$.

**V-construction.** In V-construction, each pixel is mapped to a vertex, with the pixel intensity serving as the filtration value of the corresponding vertex. Thus, we first explore the scenario where $\sigma$ is a vertex, and then extend our analysis to higher dimensional cubes.

(i) Assume $\sigma$ is a vertex, i.e., $\dim(\sigma) = 0$. Let $j \in \mathcal{I}$ be the index of the pixel in $x$ that corresponds to $\sigma$. Then, $f(\sigma) = x_j$ and thus,

$$\frac{\partial f(\sigma)}{\partial x_i} = \begin{cases} 1, & \text{if } i = j \\ 0, & \text{otherwise} \end{cases}$$

for all $i \in \mathcal{I}$.

(ii) Assume $\sigma$ is an edge, i.e., $\dim(\sigma) = 1$. Recall that $f(\sigma)$ is assigned as the maximum filtration value of its neighboring vertices, which in turn is equivalent to the maximum pixel intensity of the pixels corresponding to those neighboring vertices. Thus, we can identify the pixel associated with $\sigma$ by

    1. find neighboring vertices of $\sigma$

    2. determine the neighboring vertex with maximum filtration value

    3. identify the pixel that corresponds to the vertex found in (2)

In step 2, multiple neighboring vertices may have the same maximum filtration value. In this case, we identify the set of pixels that corresponds to all such vertices. Letting $J \subset \mathcal{I}$ denote an index set labeling the members of such set of pixels,

$$\frac{\partial f(\sigma)}{\partial x_i} = \begin{cases} 1/|J|, & \text{if } i \in J \\ 0, & \text{otherwise} \end{cases}$$

for all $i \in \mathcal{I}$. Observe that when multiple pixels contribute to $\sigma$, we distribute the gradient evenly between those pixels.

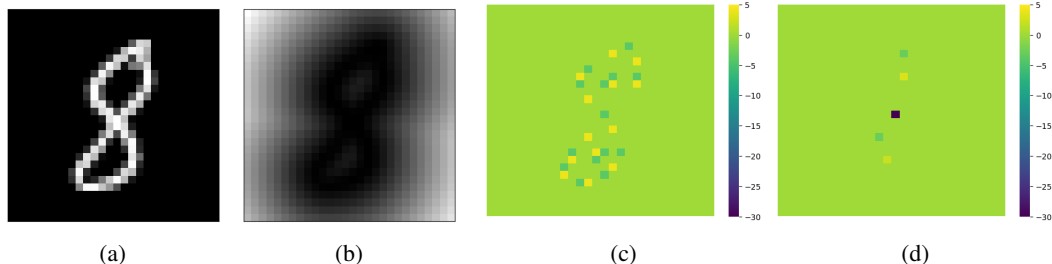

(a) (b) (c) (d)

Figure 5: (b) is the DTM transformation of (a) using $m_0 = 0.05$ on a $28 \times 28$ unit grid. (c) & (d) visualize the respective gradients of ECC and persistence landscape with respect to (b). The gradient of ECC provides more detailed and interpretable information compared to the gradient of persistence landscapes, which is very sparse.

(iii) Assume $\sigma$ is a square, i.e., $\dim(\sigma) = 2$. Recall that $f(\sigma)$ is assigned as the maximum filtration value of its neighboring edges. Therefore, once we find the neighboring edge(s) with maximum filtration value, we can repeat the process in (ii) to identify the set of pixels associated with $\sigma$. Letting $J \subset \mathcal{I}$ denote an index set labeling the members of such set of pixels,

$$\frac{\partial f(\sigma)}{\partial x_i} = \begin{cases} 1/|J|, & \text{if } i \in J \\ 0, & \text{otherwise} \end{cases}$$

for all $i \in \mathcal{I}$.

## D  APPROXIMATION OF GRADIENTS

For more detailed theoretical analysis of the approximations of the gradients, we suppose that the algorithm is to approximate the gradients of the indicator function $\mathbb{I}(f_\sigma \leq t)$ with respect to $f_\sigma$ on a fixed grid $t \in tseq = \{t_1, \ldots, t_v\}$, with $\Delta t := \frac{t_{i+1} - t_i}{2}$ being equal. Suppose the algorithm outputs approximations of gradients $\frac{\partial \mathbb{I}(f_\sigma \leq t)}{\partial f_\sigma}|_{t=t_1,\ldots,t_v}$ as $g_1, \ldots, g_v$, we treat that the gradient $\frac{\partial \mathbb{I}(f_\sigma \leq t)}{\partial f_\sigma}$ at $t$ is approximated as $g_1$ on $t \in [t_1 - \Delta t, t_1 + \Delta t)$, $g_2$ on $t \in [t_2 - \Delta t, t_2 + \Delta t)$, and $g_v$ on $t \in [t_v - \Delta t, t_v + \Delta t)$, where $g_j$ can depend on $f_\sigma$. Hence the corresponding approximation $g : [t_1 - \Delta t, t_v + \Delta t)$ is

$$g(t) = g_j, \qquad \text{for } t \in [t_j - \Delta t, t_j + \Delta t).$$

If $g$ is a good approximation of $\frac{\partial \mathbb{I}(f_\sigma \leq t)}{\partial f_\sigma}$, then

$$\int g(t) df_\sigma \approx \int \frac{\partial \mathbb{I}(f_\sigma \leq t)}{\partial f_\sigma} df_\sigma \qquad \text{for each } t \in (t_1 - \Delta t, t_v + \Delta t),$$

and

$$\int g(t) dt \approx \int \frac{\partial \mathbb{I}(f_\sigma \leq t)}{\partial f_\sigma} dt \qquad \text{for each } f_\sigma \in (t_1 - \Delta t, t_v + \Delta t).$$

We will analyze the approximations of the gradients based on these criteria.

For given $f_\sigma \in (t_1 - \Delta t, t_v + \Delta t)$, let the sigmoid approximation be $S'_{\lambda, f_\sigma} : [t_1 - \Delta t, t_v + \Delta t)$ as

$$S'_{\lambda, f_\sigma}(t) = (S'^{tseq}_{\lambda, f_\sigma})_j = -\lambda \cdot S(\lambda(t_j - f_\sigma)) \left[1 - S(\lambda(t_j - f_\sigma))\right], \qquad \text{for } t \in [t_j - \Delta t, t_j + \Delta t).$$

Similarly, let our gradient approximation be $\hat{\delta}_{\beta, f_\sigma} : [t_1 - \Delta t, t_v + \Delta t)$ as

$$\hat{\delta}_{\beta, f_\sigma}(t) = \left(\hat{\delta}^{tseq}_{\beta, f_\sigma}\right)_j = \begin{cases} -\frac{1}{\beta\sqrt{\pi}}, & \text{if } f_\sigma \in [t_{j-1}, t_j), \\ 0, & \text{otherwise}, \end{cases} \qquad \text{for } t \in [t_j - \Delta t, t_j + \Delta t).$$

**Proposition D.1.** *For all* $t \in (t_1 - \Delta t, t_v + \Delta t)$,

$$\left| \int_{t_1 - \Delta t}^{t_v + \Delta t} \hat{S}'_{\lambda, f_\sigma}(t) df_\sigma - \int_{t_1 - \Delta t}^{t_v + \Delta t} \frac{\partial \mathbb{I}(f_\sigma \leq t)}{\partial f_\sigma} df_\sigma \right| \geq 2S(-\lambda(2v - 1)\Delta t). \tag{5}$$

*And if $f_\sigma = t_j - \Delta t$ for some $j$, then*

$$\left| \int_{t_1-\Delta t}^{t_v+\Delta t} \hat{S}'_{\lambda,f_\sigma}(t)dt - \int_{t_1-\Delta t}^{t_v+\Delta t} \frac{\partial \mathbb{I}(f_\sigma \le t)}{\partial f_\sigma} dt \right| \ge |1 - 2v\lambda\Delta t \exp(-\lambda\Delta t)| . \tag{6}$$

**Proposition D.2.** *For all $t \in (t_1 - \Delta t, t_v + \Delta t)$,*

$$\int_{t_1-\Delta t}^{t_v+\Delta t} \hat{\delta}_{\beta,f_\sigma}(t)df_\sigma - \int_{t_1-\Delta t}^{t_v+\Delta t} \frac{\partial \mathbb{I}(f_\sigma \le t)}{\partial f_\sigma} df_\sigma = -\frac{2\Delta t}{\beta\sqrt{\pi}} + 1,$$

*and for all $f_\sigma \in (t_1 - \Delta t, t_v + \Delta t)$,*

$$\int_{t_1-\Delta t}^{t_v+\Delta t} \hat{\delta}_{\beta,f_\sigma}(t)dt - \int_{t_1-\Delta t}^{t_v+\Delta t} \frac{\partial \mathbb{I}(f_\sigma \le t)}{\partial f_\sigma} dt = -\frac{2\Delta t}{\beta\sqrt{\pi}} + 1.$$

Suppose the grid is fixed, so $\Delta t$ and $v$ is fixed. Then for equation 5 to go to 0, $\lambda \to \infty$ should hold. However, as $\lambda \to \infty$, the lower bound of equation 6 converges to 1, which means that the integral of the sigmoid approximation $\int_{t_1-\Delta t}^{t_v+\Delta t} \hat{S}'_{\lambda,f_\sigma}(t)dt$ becomes inconsistent. This is already expected from the vanishing gradient behavior. However, Proposition equation D.2 suggests that when $\beta$ is appropriately chosen as $\beta = \frac{\sqrt{\pi}}{2\Delta t}$, the gradient approximation becomes consistent for the integral with respect to both $f_\sigma$ and $t$.

# E    PROOFS FOR SECTION 4

*Proof for Proposition 4.1.* First, note that the sigmoid function $S(x) = \frac{1}{1+\exp(-x)}$ satisfies

$$\frac{dS}{dx}(x) = \frac{\exp(-x)}{(1+\exp(-x))^2} = S(x)(1 - S(x)),$$

and hence

$$\frac{\partial S(\lambda(t - f_\sigma))}{\partial f_\sigma} = -\lambda S(\lambda(t - f_\sigma))(1 - S(\lambda(t - f_\sigma)).$$

Since $x \mapsto |x(1-x)|$ on $[0,1]$ is maximized when $x = \frac{1}{2}$, so

$$\left\| \frac{\partial S(\lambda(t - f_\sigma))}{\partial f_\sigma} \right\|_\infty = \frac{\lambda}{4}.$$

Meanwhile,

$$\left| \frac{\partial S(\lambda(t - f_\sigma))}{\partial f_\sigma} |_{t=t_j} \right| = \lambda S(\lambda(t_j - f_\sigma)) \left[ 1 - S(\lambda(t_j - f_\sigma)) \right]$$

$$\le \lambda S(\lambda d(f_\sigma, tseq)) \left[ 1 - S(\lambda d(f_\sigma, tseq)) \right].$$

Hence

$$\left\| S'^{tseq}_{\lambda,f_\sigma} \right\|_\infty = \left\| \frac{\partial S(\lambda(t - f_\sigma))}{\partial f_\sigma} |_{t=t_1,\dots,t_v} \right\|_\infty$$

$$\le \lambda S(\lambda d(f_\sigma, tseq)) \left[ 1 - S(\lambda d(f_\sigma, tseq)) \right].$$

Hence, if $f_\sigma = t_j - \Delta t$ for some $j$, then

$$\left\| S'^{tseq}_{\lambda,f_\sigma} \right\|_\infty \le \lambda S(\lambda\Delta t) \left[ 1 - S(\lambda\Delta t) \right].$$

$$\le \lambda \exp(-\lambda\Delta t),$$

and therefore when $\lambda \exp(-\lambda\Delta t) \to 0$,

$$\inf_{f_\sigma \in [t_1-\Delta t, t_v+\Delta t)} \left\| S'^{tseq}_{\lambda,f_\sigma} \right\|_\infty \to 0.$$

$\square$

*Proof for Proposition 4.2.* $\hat{\delta}^{tseq}_{\beta,f_\sigma}$ always has the form

$$\left(0,\ldots,0,-\frac{1}{\beta\sqrt{2\pi}},0,\ldots,0\right).$$

Therefore,

$$\left\|\hat{\delta}^{tseq}_{\beta,f_\sigma}\right\|_\infty = \frac{1}{\beta\sqrt{2\pi}}.$$

$\square$

## F    PROOFS FOR APPENDIX D

Before beginning the proofs of Appendix D, we would first like to emphasize that for all $t \in (t_1 - \Delta t, t_v + \Delta t)$,

$$\int_{t_1-\Delta t}^{t_v+\Delta t} \frac{\partial \mathbb{I}(f_\sigma \leq t)}{\partial f_\sigma} df_\sigma = \mathbb{I}(t_v + \Delta t \leq t) - \mathbb{I}(t_1 - \Delta t \leq t) = -1,$$

and for all $f_\sigma \in (t_1 - \Delta t, t_v + \Delta t)$,

$$\int \frac{\partial \mathbb{I}(f_\sigma \leq t)}{\partial f_\sigma} dt = -\int \frac{\partial \mathbb{I}(f_\sigma \leq t)}{\partial t} dt = -\mathbb{I}(f_\sigma \leq t_v + \Delta t) + \mathbb{I}(f_\sigma \leq t_1 - \Delta t) = -1.$$

*Proof for Proposition D.1.* For given $t \in (t_1 - \Delta t, t_v + \Delta t)$, let $t_j \in tseq$ be such that $t \in [t_j - \Delta t, t_j + \Delta t)$. Then

$$\int \hat{S}'_{\lambda,f_\sigma}(t)df_\sigma = \int_{t_1-\Delta t}^{t_v+\Delta t} \frac{\partial S(\lambda(t_j - f_\sigma))}{\partial f_\sigma} df_\sigma$$
$$= S(\lambda(t_j - t_v - \Delta t)) - S(\lambda(t_j - t_1 + \Delta t)).$$

This is minimized when $t_j$ is close to $\frac{t_1+t_v}{2}$. Hence,

$$\int \hat{S}'_{\lambda,f_\sigma}(t)df_\sigma \geq S(-\lambda((2v-1)\Delta t) - S(\lambda((2v-1)\Delta t),$$

and hence

$$\left|\int \hat{S}'_{\lambda,f_\sigma}(t)df_\sigma - \int \frac{\partial \mathbb{I}(f_\sigma \leq t)}{\partial f_\sigma} df_\sigma\right| \geq 1 - S(\lambda(2v-1)\Delta t) + S(-\lambda(2v-1)\Delta t)$$
$$= 2S(-\lambda(2v-1)\Delta t).$$

Also, note that from the calculation in the proof of Proposition 4.1, if $f_\sigma = t_j - \Delta t$ for some $j$, then

$$\left\|S'^{tseq}_{\lambda,f_\sigma}\right\|_\infty \leq \lambda S(\lambda\Delta t)\left[1 - S(\lambda\Delta t)\right].$$
$$\leq \lambda \exp(-\lambda\Delta t).$$

And

$$\left|\int \hat{S}'_{\lambda,f_\sigma}(t)dt\right| \leq \left\|S'^{tseq}_{\lambda,f_\sigma}\right\|_\infty 2\Delta tv \leq 2\lambda\Delta tv \exp(-\lambda\Delta t).$$

Therefore,

$$\left|\int \hat{S}'_{\lambda,f_\sigma}(t)dt - \int \frac{\partial \mathbb{I}(f_\sigma \leq t)}{\partial f_\sigma} dt\right| \geq |1 - 2\lambda\Delta tv \exp(-\lambda\Delta t)|.$$

$\square$

*Proof for Proposition D.2.* For given $t \in (t_1 - \Delta t, t_v + \Delta t)$, let $t_j \in tseq$ be such that $t \in [t_j - \Delta t, t_j + \Delta t)$. Then $\hat{\delta}_{\beta, f_\sigma}(t)$ is nonzero if and only if $f_\sigma \in [t_{j-1}, t_j)$, and hence

$$\int \hat{\delta}_{\beta, f_\sigma}(t) df_\sigma = \int_{t_{j-1}}^{t_j} -\frac{1}{\beta\sqrt{\pi}} df_\sigma = -\frac{2\Delta t}{\beta\sqrt{\pi}}.$$

And for given $f_\sigma \in (t_1 - \Delta t, t_v + \Delta t)$, let $t_j \in tseq$ be such that $f_\sigma \in [t_{j-1}, t_j)$. Then $\hat{\delta}_{\beta, f_\sigma}(t)$ is nonzero if and only if $t \in [t_j - \Delta t, t_j + \Delta t)$, and hence

$$\int \hat{\delta}_{\beta, f_\sigma}(t) dt = \int_{t_j - \Delta t}^{t_j + \Delta t} -\frac{1}{\beta\sqrt{\pi}} dt = -\frac{2\Delta t}{\beta\sqrt{\pi}}.$$

$\square$

# G PROOFS FOR SECTION 5

*Proof for Proposition 5.1.* Since $\mathcal{O}_\theta(X) = g_\theta(\mathcal{C}_X(tseq))$ and $\mathcal{O}_\theta(X') = g_\theta(\mathcal{C}_{X'}(tseq))$,

$$\|\mathcal{O}_\theta(X) - \mathcal{O}_\theta(X')\|_1 = \|g_\theta(\mathcal{C}_X(tseq)) - g_\theta(\mathcal{C}_{X'}(tseq))\|_1$$
$$\leq L \|\mathcal{C}_X(tseq) - \mathcal{C}_{X'}(tseq)\|_1.$$

Now, note that ECC $\mathcal{C}_X$ can be expanded using persistence diagrams $\{\mathcal{D}_k(X) : k \geq 0\}$ as follows: if $\mathcal{D}_k(X) = \{(b_{ki}, d_{ki}) : 1 \leq i \leq n_k\}$, then

$$\mathcal{C}_X(t) = \sum_{k=0}^{\infty} (-1)^k \mathbb{I}(b_{ki} \leq t < d_{ki}).$$

Since $b_{ki}, d_{ki} \in \{t_i^*\}$, there exists $a_1, \ldots, a_m \in \mathbb{Z}$ and $n_1 < \cdots < n_{2m}$ such that $\mathcal{C}_X(t) - \mathcal{C}_{X'}(t)$ can be expressed as

$$\mathcal{C}_X(t) - \mathcal{C}_{X'}(t) = \sum_{i=1}^{m} a_i \mathbb{I}(t_{n_{2i}}^* \leq t < t_{n_{2i+1}}^*).$$

Then $\|\mathcal{C}_X(tseq) - \mathcal{C}_{X'}(tseq)\|_1$ and $\|\mathcal{C}_X - \mathcal{C}_{X'}\|_1$ is expanded as

$$\|\mathcal{C}_X(tseq) - \mathcal{C}_{X'}(tseq)\|_1 = \sum_{j=1}^{v} \sum_{i=1}^{m} |a_i| \, \mathbb{I}(t_{n_{2i}}^* \leq t_j < t_{n_{2i+1}}^*)$$

and

$$\|\mathcal{C}_X - \mathcal{C}_{X'}\|_1 = \sum_{i=1}^{m} |a_i| \, (t_{n_{2i+1}}^* - t_{n_{2i}}^*).$$

Now for each $i = 1, \ldots, m$, $\sum_{j=1}^{v} \mathbb{I}(t_{n_{2i}}^* \leq t_j < t_{n_{2i+1}}^*)$ is the number of $t_j$'s that falls within the interval $[t_{n_{2i}}^*, t_{n_{2i+1}}^*)$. But since $t_{j+1} - t_j \geq \Delta t$, such number is at most $\left\lceil \frac{(t_{n_{2i+1}}^* - t_{n_{2i}}^*)}{\Delta t} \right\rceil$, and also from $t_{n_{2i+1}}^* - t_{n_{2i}}^* \geq \Delta t$,

$$\sum_{j=1}^{v} \mathbb{I}(t_{n_{2i}}^* \leq t_j < t_{n_{2i+1}}^*) \leq \left\lceil \frac{(t_{n_{2i+1}}^* - t_{n_{2i}}^*)}{\Delta t} \right\rceil \leq \frac{2(t_{n_{2i+1}}^* - t_{n_{2i}}^*)}{\Delta t}.$$

Hence $\|\mathcal{C}_X(tseq) - \mathcal{C}_{X'}(tseq)\|_1$ can be correspondingly upper bounded as

$$\|\mathcal{C}_X(tseq) - \mathcal{C}_{X'}(tseq)\|_1 = \sum_{i=1}^{m} |a_i| \sum_{j=1}^{v} \mathbb{I}(t_{n_{2i}}^* \leq t_j \leq t_{n_{2i+1}}^*)$$

$$\leq \frac{2}{\Delta t} \sum_{i=1}^{m} |a_i| \, (t_{n_{2i+1}}^* - t_{n_{2i}}^*)$$

$$= \frac{2}{\Delta t} \|\mathcal{C}_X - \mathcal{C}_{X'}\|_1.$$

And correspondingly,

$$\|\mathcal{O}_\theta(X) - \mathcal{O}_\theta(X')\|_1 \le \frac{2L}{\Delta t} \|\mathcal{C}_X - \mathcal{C}_{X'}\|_1 .$$

$\square$

When $K = K'$, $\|f_X - f_{X'}\|_\infty = \sup_{\sigma \in K} |f_X(\sigma) - f_{X'}(\sigma)|$, and if $K \ne K'$, $\|f_X - f_{X'}\|_\infty$ is not well defined. However, there is a general distance between two filtration functions $f_X$ and $f_{X'}$ even when base simplicial complexes (or cubical complexes) are different; it is the interleaving distance $d_I(f_X, f_{X'})$. For the definition, see Section 5.1 from Chazal et al. (2016a). When $K = K'$, there is a bound

$$d_I(f_X, f_{X'}) \le \|f_X - f_{X'}\|_\infty .$$

Hence we have a general version of Theorem 5.3 as follows.

**Theorem G.1.** *Suppose $K$ is a finite simplicial complex or cubical complex. Then there exists a constant $C_K$ only depending on $K$ such that*

$$\|\mathcal{C}_X - \mathcal{C}_{X'}\|_1 \le C_K d_I(f_X, f_{X'}).$$

Proof for Theorem G.1 is in a similar manner from the proof of Wasserstein Stability Theorem of Cohen-Steiner et al. (2010).

*Proof for Theorem G.1.* From Proposition 5.2, it is sufficient to show that there exists $C'_K$ depending only on $K$ such that

$$\sum_{k=0}^\infty W_1(\mathcal{D}_k(X), \mathcal{D}_k(X')) \le C'_K d_I(f_X, f_{X'}).$$

Fix $k \ge 0$, and let $\epsilon_k := W_\infty(\mathcal{D}_k(X), \mathcal{D}_k(X'))$ be the bottleneck distance between two diagrams $\mathcal{D}_k(X)$ and $\mathcal{D}_k(X')$. Let $\gamma_k : \mathcal{D}_k(X) \to \mathcal{D}_k(X')$ be the bijection that realizes the bottleneck distance, i.e., for any $p \in \mathcal{D}_k(X)$,

$$\|p - \gamma_k(p)\|_\infty \le \epsilon_k.$$

Then 1-Wasserstein distance $W_1(\mathcal{D}_k(X), \mathcal{D}_k(X'))$ satisfies

$$W_1(\mathcal{D}_k(X), \mathcal{D}_k(X')) = \inf_\gamma \sum_{x \in \mathcal{D}_k(X)} \|x - \gamma(x)\|_\infty$$

$$\le \sum_{x \in \mathcal{D}_k(X)} \|x - \gamma_k(x)\|_\infty$$

$$\le \epsilon_k |\mathcal{D}_k(X)| .$$

And hence if we let $\epsilon := \sup_{k \ge 0}\{\epsilon_k\}$, then summing over $k \ge 0$ gives

$$\sum_{k=0}^\infty W_1(\mathcal{D}_k(X), \mathcal{D}_k(X')) \le \sum_{k=0}^\infty \epsilon_k |\mathcal{D}_k(X)|$$

$$\le \epsilon \sum_{k=0}^\infty |\mathcal{D}_k(X)| .$$

Now $\sum_{k=0}^\infty |\mathcal{D}_k(X)|$ is the number of points in persistence diagrams of all homological dimensions on $K$. This can be bounded by some constant $C'_K$ that depends only on $K$: one rough bound can be as $|\{\sigma : \sigma \in K\}|^2$, since each point in persistence diagrams has a unique pair $(\sigma_b, \sigma_d)$ of a birth simplex $\sigma_b$ and a death simplex $\sigma_d$. And therefore,

$$\sum_{k=0}^\infty W_1(\mathcal{D}_k(X), \mathcal{D}_k(X')) \le C'_K \epsilon.$$

Now from $f_X$ and $f_{X'}$ being on a finite simplicial complex $K$, they are $q$-tame (see Section 3.8 from Chazal et al. (2016a)). So from the bottleneck stability theorem (see e.g., Section 5.1 and Theorem 5.23 from Chazal et al. (2016a)), for all $k \ge 0$,

$$\epsilon_k \le d_I(f_X, f_{X'}).$$

And hence,

$$\sum_{k=0}^{\infty} W_1(\mathcal{D}_k(X), \mathcal{D}_k(X')) \leq C_K' d_I(f_X, f_{X'}).$$

$\square$

Before proving Corollary 5.4, we explain what an empirical distribution and a 'restriction of a DTM function' mean. We say $P_X$ is an empirical distribution on $X = \{X_1, \ldots, X_n\}$, when $P_X = \frac{1}{n}\sum_{i=1}^{n} \delta_{X_i}$, where $\delta_{X_i}$ is a Dirac measure on $X_i$, i.e., $\delta_{X_i}(A) = \mathbb{I}(X_i \in A)$. And suppose $\{\sigma_i\} \subset K$ be vertices of $K$ for V-construction, or top dimensional cells of $K$ for T-construction. Then we say $f_X$ is a 'restrictions of a DTM function' $d_{P_X, m_0}$, if there exists a grid $\mathcal{G} = \{x_i\} \subset \mathbb{R}^d$, with $f_X(\sigma_i) = d_{P_X, m_0}(x_i)$.

*Proof for Corollary 5.4.* From Theorem 5.3,

$$\|\mathcal{C}_X - \mathcal{C}_{X'}\|_1 \leq C_K \|f_X - f_{X'}\|_\infty.$$

Now we further bound $\|f_X - f_{X'}\|_\infty$. Note that

$$\begin{aligned}
\|f_X - f_{X'}\|_\infty &= \max_{\sigma \in K} |f_X(\sigma) - f_{X'}(\sigma)| \\
&= \max_{x \in \mathcal{G}} \left|d_{P_X, m_0}(x) - d_{P_{X'}, m_0}(x)\right| \\
&\leq \left\|d_{P_X, m_0} - d_{P_{X'}, m_0}\right\|_\infty.
\end{aligned}$$

And from Chazal et al. (2011)[Theorem 3.5],

$$\left\|d_{P_X, m_0} - d_{P_{X'}, m_0}\right\|_\infty \leq \frac{1}{\sqrt{m_0}} W_2(P_X, P_{X'}).$$

Hence putting these things together gives

$$\|\mathcal{C}_X - \mathcal{C}_{X'}\|_1 \leq \frac{(d+1)DC_K}{\sqrt{m_0}} W_2(P_X, P_{X'}).$$

$\square$

Let $d_{GH}$ be the Gromov-Hausdorff distance.

**Corollary G.2.** *Suppose $f_X$, $f_{X'}$ are Vietoris-Rips filtrations of $X$ and $X'$, respectively. Then*

$$\|\mathcal{C}_X - \mathcal{C}_{X'}\|_1 \leq C_K d_{GH}(X, X').$$

*Proof for Corollary 5.4.* From Theorem G.1,

$$\|\mathcal{C}_X - \mathcal{C}_{X'}\|_1 \leq C_K d_I(f_X, f_{X'}).$$

Then from Lemma 4.3 of Chazal et al. (2014),

$$d_I(f_X, f_{X'}) \leq d_{GH}(X, X').$$

Hence putting these things together gives

$$\|\mathcal{C}_X - \mathcal{C}_{X'}\|_1 \leq C_K d_{GH}(X, X').$$

$\square$

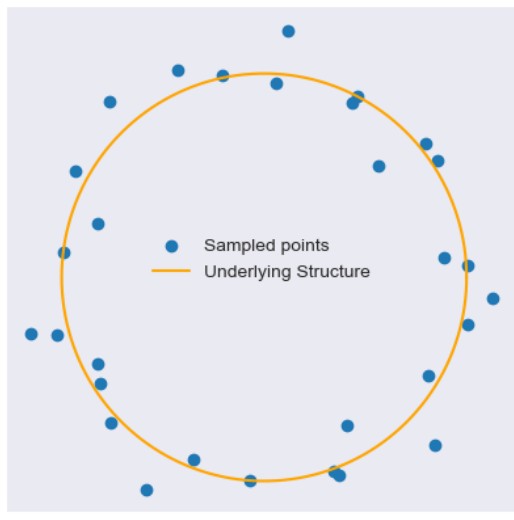

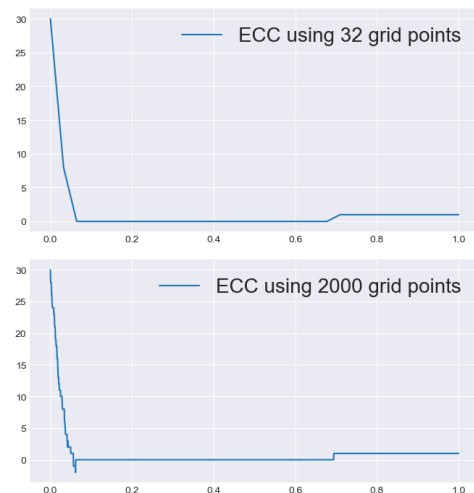

(a) 30 sampled points from unit circle with small noise.

(b) ECC of (a) calculated using Alpha filtration with 32 and 2000 grid points.

Figure 6: Within the interval $[0.06, 0.7]$, both ECCs capture the Euler characteristic of the underlying loop structure, which is 0. The ECC with 2000 grid points exhibits more noise than the ECC with 32 grid points, as the dense discretization captures even the small (noise) generators that do not represent the global structure of data.

## H CHOICE OF TDA HYPERPARAMETERS

In this section, we discuss the choice of several TDA hyperparameters in `ECLayr`: filtration, $T_{min}$, $T_{max}$, $v$, and $\beta$.

**Choice of filtration.** Although numerous filtration options exist, certain filtrations are commonly favored for specific data modalities and training contexts. For example, Vietoris-Rips and Alpha filtrations are extensively utilized for point clouds, while a sub/superlevel filtration on a filtered cubical complex is a natural choice for data with grid structure. DTM filtration provides robustness against outliers, and thereby preferable in scenarios of data contamination. Despite not being discussed here, other choices of filtrations are also available. Nevertheless, the fundamental idea remains the same; choose a filtration that can mostly effectively extract topological features from the given data.

**Choice of $[T_{min}, T_{max}]$.** A naive and convenient approach is to assign $T_{min}$ and $T_{max}$ as the minimum and maximum of possible filtration values, respectively. For instance, one can set $T_{min} = 0$ and $T_{max} = 1$ for image data with min-max normalized pixel values. An alternative method is to select $[T_{min}, T_{max}]$ as a tighter interval within the range of possible filtration values, focusing on regions of the filtration that contain meaningful topological and geometrical information. Such an interval can be identified via hyperparameter search, or chosen intuitively by examining the ECC computed for some data. For example, the ECCs in Figure 6-(b) reveal that $[0.06, 0.7]$ is a suitable interval for capturing the underlying loop structure depicted in Figure 6-(a).

**Choice of $v$.** Spacing between grid points is important as our vectorized ECC does not account for cycles that are born and dead between $t_i$ and $t_{i+1}$, where $t_i, t_{i+1} \in tseq$. Provided that the discretization is not overly sparse, the uncaptured cycles are often small (noise) generators with life span shorter than $t_{i+1} - t_i$. This implies that with appropriate discretization, noise can be partially filtered by design. Therefore, using a highly dense discretization is not necessarily beneficial, as it captures even the small (noise) generators (see Figure 6-(b)). Conversely, using an excessively sparse discretization may jeopardize the capturing of essential global features. The optimal choice of $v$ is not always evident; we recommend hyperparameter search using cross validation to determine the adequate $v$ that balances the two circumstances.

**Choice of $\beta$.** The hyperparameter $\beta$ regulates the gradient's magnitude, with smaller values of $\beta$ yielding larger gradients. However, the optimal choice of $\beta$ is somewhat ambiguous. Unfortunately,

we do not have a clear rationale for choosing $\beta$; it is contingent upon numerous factors, including model architecture and the specific task at hand. Therefore, we recommend conducting hyperparameter search via cross validation to select an appropriate $\beta$.

# I    EXPERIMENT DETAILS

All experiments were conducted with NVIDIA RTX A6000 GPU, with the exception of empirical time complexity analysis, which was run on Apple M2.

## I.1    COMPUTATIONAL EFFICIENCY

The runtime metrics are computed on the training set of MNIST, Br35H, and synthetic data. MNIST contains 60000 training samples of size $28 \times 28$. For Br35H, we use the same training set as the experiment conducted in Section 6.4, which is 209 samples of size $112 \times 112$. We generate 1000 samples of size $224 \times 224$ for the synthetic data, where each pixel is randomly sampled from a uniform distribution. Both ECC and PH use superlevel set filtration on V-constructed cubical complex. PH is computed using the GUDHI package, while Algorithm 1 is used to compute ECC with $v = 32$ on interval $[0, 1]$.

## I.2    TOPOLOGICAL AUTOENCODER

The encoder and decoder network each consists of three fully connected layers, with input dimension size 101, hidden dimension size 32, and latent dimension size 2. BatchNorm and ReLu nonlinearity is used after each layer, except for the latent dimension. We place one `ECLayr` at the beginning of the encoder and one `ECLayr` at the latent dimension in order to compute the MAE loss between ECC of input point cloud and ECC of latent representation. This MAE loss acts as a topology regularizing term, with lambda=0.001 controlling its magnitude. Vietoris-Rips filtration is employed to compute ECC and max dimension is restricted to 1. $v$ is set to 1000 over interval $[0, 2]$, where filtration values indicate edge length. $\beta$, which controls the magnitude of the gradient, is assigned as 0.01 and we do not use $g_\theta$ for this experiment. Adam optimizer is used for training with batch size 32 and learning rate 0.0001. We run for 100 epochs and adopt early stopping after patience 10.

## I.3    MNIST DATASET

The MNIST dataset contains 60000 training data and 10000 test data of handwritten digits from 0 to 9. We implement a 4 layer baseline model, consisting of two CNN layers followed by two fully connected layers, with ReLU nonlinearity between every layer. Both CNN layers use $3 \times 3$ kernels with stride 1 and padding 1. Each CNN layer has channel size 32 and 1, respectively. The output of the CNN network is flattened and passed to the subsequent fully connected layers with hidden dimension of 64. For training, we employ the Adam optimizer with learning rate 0.001 and batch size 32. The learning rate is decayed by a factor of 0.1 when the validation loss plateaus for 10 epochs. While we use a maximum of 1000 epochs for training, early stopping is implemented to stop training after the validation loss plateaus for 25 epochs. Cross-entropy loss is used for classification. We compare the performance of our proposed layer with the base model, and two other topological layers applicable to image datasets: PersLay (Carrière et al., 2020) and PLLay (Kim et al., 2020). Utilizing a very simple model on limited training samples, we observed random failures across all models with outliers significantly affecting the outcome. To remove the influence of outliers and solely evaluate model performance, we repeat each experiment 15 times and select the top 10 test accuracies for assessment. 30% of the training data is used as a validation set and model performance is evaluated on the complete test data.

**Data Scarcity.** To observe how model performance changes with data size, we sample training data of size $100, 300, 500, 700$ and $1000$ with equal proportion for each label. For all topological layers, we place a parallel layer at the beginning of the network (referred to as CNN + EC(i), CNN + Pers, and CNN + PL(i)). For `ECLayr` and PLLay, which allow backpropagation, we add an additional layer after the last convolutional layer (referred to as CNN + EC, CNN + DECT and CNN + PL). $\beta = 0.01$ is used to control the gradient intensity for `ECLayr`. For all topological layers, we implement superlevel set filtrations on T-constructed cubical complex and use $v = 32$

| Models | Data Size | | | | |
|---|---|---|---|---|---|
| | 100 | 300 | 500 | 700 | 1000 |
| CNN | 69.075 ($\pm$1.049) | 83.282 ($\pm$0.566) | 85.346 ($\pm$0.305) | 86.868 ($\pm$0.397) | 89.066 ($\pm$0.518) |
| CNN + EC(i) | 73.417 ($\pm$0.718) | 86.549 ($\pm$1.050) | 87.649 ($\pm$0.251) | 88.642 ($\pm$0.335) | **90.659** ($\pm$0.397) |
| CNN + EC | **73.652** ($\pm$0.983) | **86.551** ($\pm$1.035) | **88.335** ($\pm$0.458) | **88.755** ($\pm$0.319) | 90.361 ($\pm$0.400) |
| CNN + DECT | 70.872 ($\pm$1.484) | 84.004 ($\pm$0.364) | 85.702 ($\pm$0.502) | 86.940 ($\pm$0.343) | 89.421 ($\pm$0.234) |
| CNN + Pers | 64.982 ($\pm$2.225) | 81.256 ($\pm$1.376) | 84.239 ($\pm$0.617) | 86.200 ($\pm$0.786) | 88.444 ($\pm$0.474) |
| CNN + PL(i) | 66.425 ($\pm$2.078) | 83.176 ($\pm$1.156) | 86.110 ($\pm$0.598) | 87.996 ($\pm$0.476) | 90.072 ($\pm$0.364) |
| CNN + PL | 69.656 ($\pm$2.673) | 83.382 ($\pm$1.764) | 86.251 ($\pm$1.200) | 88.123 ($\pm$0.231) | 90.164 ($\pm$0.713) |

Table 4: Test accuracy of models trained on different sizes of MNIST dataset. For each data size, the best accuracy is highlighted in bold.

| Models | Corruption & Noise Probability | | | | |
|---|---|---|---|---|---|
| | 0.00 | 0.05 | 0.10 | 0.15 | 0.20 |
| CNN | 89.066 ($\pm$0.518) | 86.556 ($\pm$0.495) | 85.711 ($\pm$0.546) | 83.288 ($\pm$0.735) | 81.357 ($\pm$0.464) |
| CNN + EC(i) | **90.659** ($\pm$0.397) | **88.315** ($\pm$0.542) | **86.844** ($\pm$0.564) | 84.341 ($\pm$0.753) | **82.122** ($\pm$0.791) |
| CNN + EC | 90.361 ($\pm$0.400) | 88.164 ($\pm$0.700) | 86.762 ($\pm$0.700) | 84.310 ($\pm$0.800) | 81.976 ($\pm$0.647) |
| CNN + Pers | 88.444 ($\pm$0.474) | 87.686 ($\pm$1.229) | 86.680 ($\pm$0.304) | **84.385** ($\pm$0.559) | 81.845 ($\pm$0.483) |
| CNN + PL(i) | 90.072 ($\pm$0.364) | 85.723 ($\pm$0.740) | 84.742 ($\pm$0.641) | 82.303 ($\pm$0.971) | 80.593 ($\pm$0.493) |
| CNN + PL | 90.164 ($\pm$0.713) | 85.665 ($\pm$0.665) | 84.9 ($\pm$0.637) | 82.668 ($\pm$0.595) | 80.852 ($\pm$0.926) |

Table 5: Test accuracy of models trained on 1000 MNIST data with different corruption & noise probability. For each corruption and noise probability, the best accuracy is highlighted in bold.

over interval $[0, 1]$. PersLay uses line point transform and a $10 \times 10$ unit grid for learnable weights over interval $[0, 1] \times [0, 1]$. $top2$ function is used as the permutation invariant operation for PLLay and PersLay. After the topological descriptor is computed from the respective layers, it is fed into a fully connected layer $g_\theta$ with output dimension of 32 to compute the final output of each topological layer. We concatenate the output of the topological layer with the output of the CNN network to feed it to the subsequent fully connected layer. See Table 4 for the full experiment results

**Data Contamination.** To observe how robust models are against data contamination, we consider a corruption and noise process where the pixels are randomly omitted and subsequently contaminated by random noise between 0 and 1 with probability $0.05, 0.1, 0.15$, and $0.2$. We apply different levels of noise to 1000 training samples with equal proportion for each label. The overall architecture and training scheme remain the same as before, except that we now place two parallel layers at the beginning of the network using DTM filtration. For each DTM filtration, we align the data on a unit grid and use $m_0 = 0.05$ and $m_0 = 0.2$ to examine the topological structure at different local/global scales. We use the interval $[0.02, 0.28]$ and $[0.06, 0.29]$ for $m_0 = 0.05$ and $m_0 = 0.2$ respectively. For CNN + EC and CNN + PL, we place an additional layer using DTM filtration with $m_0 = 0.05$ after the last convolutional layer. $\beta$ is set at 0.01. See Table 5 for the full experiment results

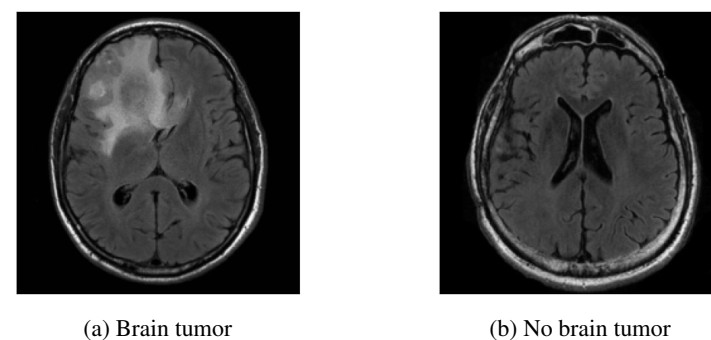

(a) Brain tumor         (b) No brain tumor

Figure 7: Example brain MRI images from Br35H dataset.

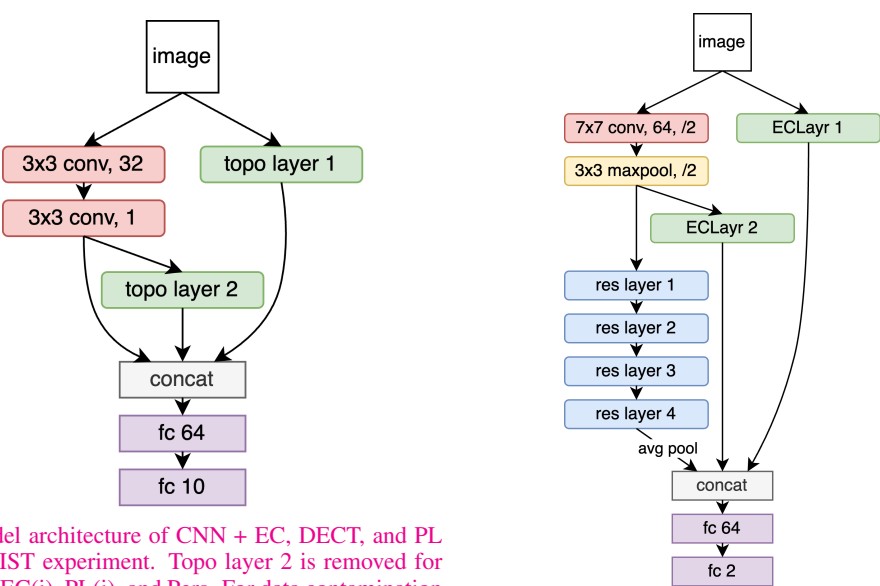

(a) Model architecture of CNN + EC, DECT, and PL for MNIST experiment. Topo layer 2 is removed for CNN + EC(i), PL(i), and Pers. For data contamination scenarios, another topological layer is employed parallel to topo layer 1.

(b) Model architecture of ResNet + EC for Br35H experiment. ECLayr 2 is removed for ResNet + EC(i).

Figure 8: Illustration of model architectures.

## I.4 BR35H DATASET

The Br35H dataset contains 3000 brain MRI images, used for a binary classification task of detecting brain tumors. There is 1500 data for each label. As images in this dataset have different size for each dimension and a varying number of channels, we preprocess the data by cropping along the shorter dimension, resizing it to $112 \times 112$, and converting it to grayscale to unify the number of channels. We implement ResNet18 (He et al., 2016) as our baseline model, with an additional fully connected layer of size 64 appended at the end of the network. For ResNet + EC(i), we add a parallel `ECLayr` at the beginning of the network and concatenate the output with the residual layer output before feeding to the fully connected layer. For ResNet + EC, we place an additional `ECLayr` before the first residual layer. We use the same training scheme as before, with only a change in learning rate and batch size. As we are using `ECLayr` in conjunction with a large model, we assign different learning rates for ResNet18 and `ECLayr`. The learning rate for `ECLayr` is set at 0.001, while 0.01 is used for ResNet. We use batch size 64. For `ECLayr` used at the beginning of the network, we use $v = 64$ over interval $[0.4, 1]$. The second `ECLayr` uses $v = 64$ over interval $[0.3, 1]$. Both layers use superlevel set filtration on V-constructed cubical complexes, and we employ a linear layer of size 64 for $g_\theta$. $\beta$ is set at 0.01. We only use $10\%$ of the data as training samples and $30\%$ of training data is used for validation.

| Filtration | | | Interval $[T_{min}, T_{max}]$ | | |
|---|---|---|---|---|---|
| T, sub | V, sup | V, sub | $[0.1, 1]$ | $[0, 0.8]$ | $[0.1, 0.8]$ |
| 88.927 ($\pm 0.262$) | 89.205 ($\pm 0.291$) | 88.817 ($\pm 0.219$) | 89.249 ($\pm 0.230$) | 89.659 ($\pm 0.430$) | 89.467 ($\pm 0.134$) |
| Discretization $v$ | | | Gradient Control $\beta$ | | |
| 16 | 64 | 128 | 0.1 | 0.001 | 0.0001 |
| 89.299 ($\pm 0.453$) | 89.271 ($\pm 0.274$) | 88.979 ($\pm 0.228$) | 89.228 ($\pm 0.176$) | 89.206 ($\pm 0.332$) | 88.659 ($\pm 0.657$) |

Table 6: Test accuracy on 1000 MNIST data for different choice of hyperparameters.

## I.5 HYPERPARAMETER INFLUENCE

`ECLayr` introduces four hyperparameters: filtration, interval $[T_{min}, T_{max}]$, discretization $v$, and gradient control $\beta$. To evaluate the influence of each hyperparameter on performance, we provide an ablation study on 1000 noiseless samples from the MNIST dataset. We vary a single hyperparameter while keeping all others consistent with the experiment in Section 6.3. The test accuracies of EC + CNN model are presented in Table 6. "T", "V", "sub", and "sup" for different choices of filtration refer to T-construction, V-construction, sublevel set filtration, and superlevel set filtration, respectively.

