# OpenReview forum: "ECLayr: Fast and Robust Topological Layer based on Differentiable Euler Characteristic Curve"
_ICLR.cc/2025/Conference — Submitted to ICLR 2025_

### Official Review · Reviewer_98ug · 2024-10-22

**Soundness:** 2
**Presentation:** 3
**Contribution:** 2
**Rating:** 5
**Confidence:** 3

**Summary:**

The paper introduces a topological data representation based on Euler characteristic curves (ECC). The representation is obtained via discritizing the ECC and then processing it with a parametrized, learnable map. To make the representation differentiable and avoid vanishing gradients, the authors approximate the derivative of the indicator function with a shifted Gaussian function. The paper highlights several advantages of this representation, including computational efficiency, mitigation of vanishing gradient issues, and stability under data perturbation. Numerical experiments are provided to demonstrate the representation’s effectiveness.

**Strengths:**

* The paper is well-structured and easy to follow.
* The proposed representation is computationally efficient and applicable to general data and filtrations, allowing a convenient integration with deep learning models for diverse tasks.
* The proposed representation mitigates the vanishing gradients issues.
* The paper includes extensive experiments, covering various application scenarios and provides an adequate comparison with existing methods.

**Weaknesses:**

* To avoid vanishing gradient, the authors approximate the (distributional) derivative of the indicator function with a Gaussian function and shift the Gaussian center to align with the filtration values. However, this approximation seems unnecessary, as the approach is equivalent to always setting $\partial \mathbb{I}(f_\sigma \leq t_i)/\partial f_\sigma$ as a fixed constant $\gamma$, where $f_\sigma \in [t_i, t_{i+1})$. This is also equivalent to approximating the indicator function $\mathbb{I}(f_\sigma \leq t_i)$ with a linear function $f_\sigma \mapsto \gamma (f_\sigma - t_i)$. It is then unclear how novel this technique is.

* Another concern with the shifting is that, as the gradient is set to a constant independent of $f_\sigma$, the information about the filtration value is lost and all simplices with filtration values between $[t_i, t_{i+1})$ have a same gradient term $\partial \mathbb{I}(f_\sigma \leq t_i)/\partial f_\sigma=\gamma$. This leads to the topology information being distorted during back propagation. As a comparison, the sigmoid approximation [1] produces a gradient of the form $\lambda S^\prime(\lambda( f_\sigma -t_i))$, which contains information about $f_\sigma$.

* Regarding the stability result, Proposition 5.1 indicates the Lipschitz constant can be bounded by $2L/\Delta t$. This bound diverges with a rate $\Theta(\Delta t^{-1})$ as the grid size $\Delta t$ tends to zero. Hence, I question if it is useful, as a smaller $\Delta t$ corresponds to a better approximation of ECC during discretization. Note in the experiments described in H.2, $1/\Delta t = 1/(2/1000) = 500$; and in H.4, $1/\Delta t = 1/(.6/64)\approx 106$. As a comparison, persistence image, which is a PH-based representation and is also obtained with discretization, is stable with a grid-size-free Lipschitz constant [2].

Overall, while the proposed representation offers computational convenience, it is less informative than PH-based representation, especially with the discretization of ECC and the shifting techniques. I agree with the author that "understanding this tradeoff (between computational efficiency and expressiveness) is essential" and would have appreciated a more intuitive and theoretical discussion on this tradeoff. For example, how one can expect the proposed representation is better than a naive representation: a vector of the top-50 largest filtration values? This naive representation also does not require PH computations and is stable as long as the filtration function is.

---

[1] Roell, Ernst, and Bastian Rieck. "Differentiable Euler characteristic transforms for shape classification."

[2] Adams, Henry, et al. "Persistence images: A stable vector representation of persistent homology."

**Questions:**

* When computing the output of ECLayr from the discretized ECC, the paper mentions that "ECC may potentially contain some noise information. Thus, ... employ a differentiable parametrized map $g_\theta$ to project ECC to a learnable task-optimal representation". How should one expect this map to filter out noise information?
* Just a reminder: the current heading in the paper is for last year’s conference.

---

> ### Author Response · Authors · 2024-11-25
>
> We appreciate your valuable comments and questions. We address each of them below.
>
> 1. **[Concern on gradient approximation]** Thank you for your thorough feedback. First, we would like to clarify that our approach is not equivalent to approximating the indicator function with a linear function $\gamma (f_\sigma - t)$. Specifically, the gradient in our method produces a constant value $\gamma$ at $t=f_\sigma$ (or at $t_{i+1}$ after shifting, given $f_\sigma \in (t_i, t_{i+1}]$) and zero elsewhere, whereas the gradient of the linear function is $\gamma$ for all values of $t$. Nevertheless, you are absolutely correct in stating that the magnitude of the gradient at $t = f_\sigma$ is always fixed to a constant $\gamma$ in our approach. While the simplicity of our method may initially appear to limit its novelty, it sufficiently addresses key issues associated with sigmoid approximation. Primarily, it resolves the gradient inconsistency and vanishing problems, as detailed in Section 4 of our paper. Additionally, our backpropagation technique facilitates the use of Algorithm 1 during the forward pass, achieving enhanced computational efficiency with a time complexity of $O(N+v)$ compared to the $O(vN)$ required by the sigmoid approximation; the $O(vN)$ time complexity of ECC computation using sigmoid approximation arises from the need to apply the sigmoid function to all $t_i \in \text{tseq}$ during each iteration across all $\sigma \in K$.
>
> 2. **[Concern on shifting]** Thank you for this insightful question. First, we would like to clarify that although the gradient at $t = f_\sigma$ is fixed to a constant $\gamma$, the information of filtration value is not lost. This information is incorporated into the backpropagation location $t_{i+1}$ through the dependence $t_i < f_\sigma \leq t_{i+1}$. Similarly, for the sigmoid approximation with a very large $\lambda$, the gradient is also fixed to a constant $\lambda/4$ at $t = f_\sigma$ and nearly zero elsewhere. In both cases, the filtration value is not encoded directly in the gradient itself but rather in the location $t = f_\sigma$ to which the gradient is backpropagated. In fact, sigmoid approximation is more prone to information loss due to potential gradient inconsistency or vanishing, where parts or even the entirety of the gradient information can be lost. Second, we completely agree with your observation regarding the distortion of topology information. However, this distortion primarily occurs during the forward pass when ECC is approximated as a vector. Specifically, the jump of the indicator $I(f_\sigma \leq t)$ is reflected at $t_{i+1}$ rather than at $t=f_\sigma$, given $t_i \leq f_\sigma \leq t_{i+1}$. This type of distortion is inherent to all vector approximation methods, including PLLay and DECT. Since we backpropagate the gradient to $t_{i+1}$, the exact location where the jump was reflected in the forward pass, we believe no additional distortion is introduced during backpropagation. Moreover, provided that the discretization is not excessively sparse, this distortion is unlikely to significantly affect the global topology of data.
>
> 3. **[Stability result]:** While smaller $\Delta t$ allows for a more accurate approximation of ECC, we would like to emphasize that using a highly dense discretization is not necessarily beneficial. Given the discretized interval, the vectorized ECCs are not able to capture cycles that are born and perished between $t_i$ and $t_{i+1}$. However, cycles with very short life spans are often insignificant (noisy) generators. Therefore, excessively dense discretization may not be beneficial, as including numerous small, noisy generators in the learning process can lead to increased variance (please refer to the illustration provided in Figure 6 of the revised manuscript). By avoiding excessively fine discretization, we can prevent overly loose stability bounds in Proposition 5.1. However, we acknowledge that the stability results of ECC are less strict compared to PH-based descriptors, such as persistence images, as noted at the end of Section 5.
>
> 4. **[Comparison with naive representations]:** This is a great question. A naive representation, such as a vector of top-50 largest filtration values, fails to preserve meaningful topological and geometrical information. For instance, the loop structure in digit 0 of the MNIST dataset is lost in such a representation. In contrast, our proposed representation captures the Euler characteristic of the underlying structure within the data, thereby effectively preserving topological and geometrical information.
>
> 5. **[Ans to Q1]:** Deep learning models are renowned for their ability to autonomously extract meaningful features from data during training. In a similar vein, we anticipate that $g_\theta$ will effectively retain relevant information while filtering out noise that may not contribute to the task.

---

> > ### Comment · Reviewer_98ug · 2024-11-26
> >
> > Thank you for the detailed responses. However, I still have a few concerns.
> >
> > **Gradient approximation:** when calculating $\partial \mathbb{1}(f_\sigma \leq t_i)/\partial f_\sigma$, $f_\sigma$ should be treated as the input variable rather than $t$. Specifically, given $f_\sigma \in (t_i, t_{i+1}]$: (1) for $j\neq i$, one has $\partial \mathbb{1}(f_\sigma \leq t_j)/\partial f_\sigma=0$ (here no approximation is used); (2) for $j=i$, the proposed method essentially replaces $f_\sigma\mapsto \mathbb{1}(f_\sigma \leq t_i)$ with $f_\sigma \mapsto \gamma(f_\sigma-t_i)$, so that $\partial \mathbb{1}(f_\sigma \leq t_i)/\partial f_\sigma\approx \gamma$. Therefore, this is still a linear approximation in my view. Besides, if there is no substantial difference between the two descriptions: "setting the derivative as a constant" and "considering distribution derivative and shifting the Gaussian center to a gridpoint", I would suggest using the former in Section 4.2, as it is more transparent and avoids unnecessary complexities. (I appreciate the comparison regarding time complexity.)
> >
> > **Shifting:** I understand the filtration value is reflected by the gridpoint $t_i$. However, my concern is that the exact location of $f_\sigma$ within the interval $[t_i, t_{i+1})$ is lost in the proposed method. That is, in the eyes of ECLayr, all simplices can only take values in {$t_1, \ldots, t_v$}.  In contrast, with Sigmoid approximation and a proper chosen (not too large) $\lambda$, the gradient term $\lambda S^{\prime}\left(\lambda\left(f_\sigma-t_i\right)\right)$ reflects the exact $f_\sigma$ value and thus the representation is capable to distinguish different filtration values that lie in one interval. I even anticipat this mechanism could help compensate for the information loss in the vectorization of ECC.
> >
> > **Expressivity:** Thank you for providing the comparison between ECLayr and the naive representation. However, as this is just an illustrative example, my primary concern remains: while the representation is faster than PH, how does it compromise on expressivity? As I mentioned in the initial review, I would greatly appreciate detailed discussion on this aspect, e.g., what information is gained compared to naive representations and what is lost compared to PH-based representations. To me, addressing this tradeoff between expressivity and computational efficiency is crucial in demonstrating the effectiveness of the proposed representation.

---

> > > ### Author Response · Authors · 2024-12-03
> > >
> > > We sincerely thank the reviewer for their time and thoughtful feedback. We would like to address each of your remaining concerns below.
> > >
> > > - **Gradient approximation:** As a function of $f_\sigma$, we agree with the reviewer's view that it is a linear approximation. We also appreciate the suggestion regarding the wording. We will make sure to reflect this on our final print to avoid confusion.
> > >
> > > - **Shifting:** Thank you for the additional feedback on this matter. As the reviewer has mentioned, the exact location of $f_\sigma \in [t_i, t_{i+1})$ is lost in our proposed method. In our opinion, however, the precise location of $f_\sigma$ within $[t_i, t_{i+1})$ is of negligible significance as long as the discretization is not excessively sparse. This is due to the fact that ECC focuses on the global structure of data, represented by a constant value of Euler characteristic that persists for an extended interval along the filtration (please refer to Figure 6 for an illustration). In contrast, shifting $f_\sigma \in [t_i, t_{i+1})$ to $t_{i+1}$ is a small local perturbation that does not significantly influence the global topology. Therefore, distinguishing different filtration values that lie in a single interval holds minimal importance in terms of capturing the global structure, and we believe the minor distortions induced by shifting are negligible.
> > >
> > > - **Expressivity:** Thank you for the elaboration on the initial review. Below, we provide a discussion on the tradeoff in expressivity between PH, ECC, and naive representations.
> > >     1. **Comparison with PH:** The compromise in expressivity stems from ECC aggregating topological features across all homology dimensions, whereas PH provides separate topological summaries for each homology dimension. As an illustrative example, consider two distinct datasets sampled from different underlying structures: (i) a set of points sampled from a single loop, and (ii) a set of points sampled from two \emph{separate} loops. PH will output signals indicating 1 connected component ($\beta_0 = 1$) and 1 loop ($\beta_1 = 1$) for the first data, and 2 connected components ($\beta_0 = 2$) and 2 loops ($\beta_1 =2$) for the second data, thereby distinctly distinguishing the two datasets. In contrast, ECC loses the detailed information of each homology dimension and preserves only the high-level aggregated information. Consequently, ECC fails to clearly distinguish the two datasets because the Euler characteristics of the underlying structures are identical. Specifically, $\beta_0 - \beta_1 = 1 - 1 = 0$ for the first case, and $\beta_0 - \beta_1 = 2 - 2 = 0$ for the second case. However, we emphasize that in certain cases, ECC can be just as informative as PH for a given task. In the MNIST dataset, for instance, ECC is sufficient for distinguishing datasets with no loops, one loop, and two loops.
> > >         1. no loops, e.g., digit 1: EC = $\beta_0 - \beta_1 = 1 - 0 = 1$
> > >         2. one loop, e.g., digit 6: EC = $\beta_0 - \beta_1 = 1 - 1 = 0$
> > >         3. two loops, e.g., digit 8: EC = $\beta_0 - \beta_1 = 1 - 2 = -1$
> > >
> > >         In this context, ECC exhibits minimal to negligible information loss compared to PH, while offering a significantly simpler and computationally efficient topological feature. Ultimately, the amount of tradeoff in expressivity between ECC and PH is contingent upon the data and the task at hand. If the topology information of each homology dimension is crucial for a given task, ECC results in significant information loss; conversely, if only the aggregated information is sufficient, ECC demonstrates minimal loss in expressivity and may be preferable due to its computational benefits.
> > >
> > >     2. **Comparison with naive representations:** Naive representations often fail to preserve the global topological/geometric information, or provide limited insight based on specific homology dimensions. In contrast, ECC offers a more comprehensive summary of the global topology of data by integrating information from all homology dimensions. Thus, ECC achieves balance between capturing global topological structure and maintaining computational efficiency.
> > >
> > > We hope the above response addresses your concerns.

---

### Official Review · Reviewer_zMtu · 2024-10-30

**Soundness:** 2
**Presentation:** 3
**Contribution:** 2
**Rating:** 5
**Confidence:** 3

**Summary:**

The paper introduces ECLayr, a computationally efficient topological layer for deep learning architectures, utilizing the Euler characteristic curve (ECC) rather than persistent homology (PH). ECLayr aims to overcome the computational limitations of PH while preserving essential topological insights. The layer is designed to integrate into various neural networks by providing stable backpropagation, addressing issues like vanishing gradients. ECLayr’s effectiveness is demonstrated through experiments in classification and topological autoencoders, highlighting its speed and robustness against data contamination.

**Strengths:**

- ECLayr presents an interesting approach to leveraging topological features by focusing on ECC instead of PH.
- The paper covers detailed theoretical evaluations of the proposed method.
- The paper is generally well-organized and explains complex concepts in topological data analysis clearly, although some explanations in the backpropagation section may benefit from further clarification.
- The proposed ECLayr layer has potential for broad application in scenarios where computational efficiency is crucial, such as high-dimensional data or real-time applications.

**Weaknesses:**

- Although the paper introduces ECC in a deep learning context, ECC itself is not a new concept, limiting the work's novelty. The paper might benefit from comparisons with recent, alternative topological descriptors beyond PH to clarify its advantages.
- While experiments cover some datasets, they primarily test standard tasks and small datasets, which might not fully showcase the layer’s potential in more complex, large-scale applications.
- Although ECC-based methods are computationally efficient, they are also inherently less robust than PH due to dependence on small generators. Further testing with different noise levels or adversarial robustness could provide a clearer view of its resilience.
- The stability and backpropagation advantages are discussed in some depth, but additional formal proofs or extended stability comparisons with PH-based methods would solidify the theoretical foundation.

**Questions:**

- The stability analysis primarily focuses on ECC robustness against small noise. Could the authors provide more comparative results showing how ECLayr performs under more significant noise compared to PH-based methods?
- Could you clarify the hyperparameters and initialization procedures in ECLayr's construction? Some details in the choice of differentiable filtrations seem crucial to replicating results.

---

> ### Author Response · Authors · 2024-11-25
>
> Thank you for your valuable comments and suggestions. We address each of them below.
>
> 1. **[Novelty of the work]:** Thank you for raising this point. While ECC itself is not a new concept, we would like to highlight that the development of a differentiable topological layer based on ECC is novel. The idea of differentiating ECC with respect to the layer input has only been previously explored in [1], which was limited to the height filtration and lacked detailed theoretical analysis. In contrast, our work introduces a general framework that supports any differentiable filtration and is accompanied by theoretical stability results. Furthermore, we propose a novel backpropagation method that addresses the gradient inconsistency issue caused by the sigmoid approximation used in [1]. By removing the need for sigmoid approximation, our approach also improves the forward pass’s time complexity to $O(N+v)$, compared to the $O(vN)$ complexity associated with the sigmoid-based method; the $O(vN)$ time complexity of ECC computation using sigmoid approximation arises from the need to apply the sigmoid function to all $t_i \in \text{tseq}$ during each iteration over all $\sigma \in K$. Consequently, our primary contribution is not simply leveraging ECC for computational efficiency, but introducing a comprehensive framework for differentiable ECC. This framework includes a novel backpropagation method and rigorous theoretical analysis of both stability and backpropagation.
>
> 2. **[Stability comparison with PH-based methods]:** At the end of Section 5, we briefly mentioned that our stability results are less strict compared to those of PH. Below, we provide a direct comparison for clarity. By combining Proposition 5.1. and 5.2., the output of our layer is bounded by
> $$
> \Vert \mathcal{O}\_{\theta}(X) - \mathcal{O}\_{\theta}(X') \Vert \_{1} \leq \frac{2L}{\Delta t} \Vert \mathcal{C}\_{X}-\mathcal{C}\_{X'} \Vert _{1} \leq  \frac{2L}{\Delta t} \cdot 2\sum\_{k=0}^{\infty}W\_{1}(\mathcal{D}\_{k}(X),\mathcal{D}\_{k}(X')).
> $$
> whereas the output of [2](Theorem 4.1.) is bounded by
> $$
> \Vert \mathcal{O}\_{\theta}(X)-\mathcal{O}\_{\theta}(X') \Vert\ _{1} \leq  L \cdot d\_{B}(\mathcal{D}(X),\mathcal{D}(X')).
> $$
> The results indicate that PH-based methods are generally bounded by the bottleneck distance, whereas our layer adopts a more relaxed bound based on the Wasserstein distance. We hope this offers a clear comparison of the stability results between ECC and PH-based methods.
>
> 3. **[Ans to Q1 and experiment on larger dataset]:** Thank you for your suggestion. In response, we conducted an additional experiment on ORBIT5K, a widely used point cloud benchmark dataset in TDA. The data contains 5000 samples, offering a relatively larger sample size compared to our previous experiments. In this experiment, we introduced increased noise levels of up to $30$%. Due to limited computational resources, the experiment was conducted only on our ECC based models. The results were on par with the numbers reported in [2] up to a 10% noise level, but showed a significant drop in performance beyond this threshold.
>
> 4. **[Ans to Q2]:** ECLayr introduces four TDA hyperparameters: filtration, interval $[T_{min}, T_{max}]$, discretization $v$, and gradient control $\beta$. The initialization procedure of ECLayr proceeds as follows. First, a filtration is constructed by selecting an appropriate simplicial complex $K$ and function $f: K \rightarrow \mathbb{R}$. Next, an interval $[T_{min}, T_{max}]$ is chosen as the domain on which ECC will be computed. This interval is then discretized into $v$ evenly-spaced grid points. The vectorized ECC can subsequently be computed using Algorithm 1. During backpropagation, $\beta$ controls the magnitude of the gradient, with smaller values of $\beta$ yielding stronger gradients. For your reference, the hyperparameter settings used in our experiments are listed below and can also be found in Experiment Details section of the Appendix.
>
> MNIST - data scarcity
>  - filtration: superlevel set filtrations on T-constructed cubical complex
> - $[T_{min}, T_{max}]$: $[0,1]$
> - $v$: 32
> - $\beta$: 0.01
>
> MNIST - data contamination
> - filtration: DTM filtrations with $m_0=0.05$ and $m_0=0.2$, respectively
> - $[T_{min}, T_{max}]$: $[0.02, 0.28]$ and $[0.06, 0.29]$ for $m_0 = 0.05$ and $m_0 = 0.2$ respectively
> - $v$: 32
>  - $\beta$: 0.01
>
> Br35H
> - filtration: superlevel set filtrations on V-constructed cubical complex
> - $[T_{min}, T_{max}]$: $[0.4, 1]$ for first ECLayr placed in parallel to ResNet and $[0.3, 1]$ for second ECLayr placed between max pool and first residual layer of ResNet.
> - $v$: 64
> - $\beta$: 0.01
>
> Roell, Ernst, and Bastian Rieck. "Differentiable Euler characteristic transforms for shape classification." [1]
>
> Kwangho Kim, Jisu Kim, Manzil Zaheer, Joon Kim, Frédéric Chazal, and Larry Wasserman. "Pllay:Efficient topological layer based on persistent landscapes". [2]

---

> > ### Comment · Reviewer_zMtu · 2024-11-27
> >
> > Thank you very much. The reviewer appreciates the authors' comment. Could the authors elaborate further on the statement: "The results were on par with the numbers reported in [2] up to a 10% noise level, but showed a significant drop in performance beyond this threshold"? Additional insights into the factors contributing to the drop in performance would be helpful.

---

> > > ### Author Response · Authors · 2024-12-03
> > >
> > > We sincerely thank the reviewer for their time and valuable comments. We would like to address each of your questions below.
> > >
> > > - **Elaboration on experiment:** Our ECC-based models demonstrated performance comparable to PLLay[2] up to $10$% noise level, even marginally outperforming PLLay by $0.38$% in the noiseless scenario, and exhibiting test accuracy difference within $0.35$% and $1.4$% for $5$% and $10$% noise levels, respectively. However, we observed a performance drop at $15$% noise level, showing a difference of about $4.5$% compared to PLLay. The performance gap gradually increased up to $9$% for noise level of $30$%. We note that this additional experiment should be regarded merely as a reference, as we did not run the experiments for PLLay and the performance metrics for PLLay were taken from [2].
> > >
> > > - **Insights into drop in performance:** We conjecture that two factors contributed to the drop in performance of ECC-based models. The first factor is the inherent instability of ECC relative to PH, as outlined in the stability analysis of our paper. The second and more important factor is the selection of the interval $[T_{min}, T_{max}]$. In our experiment, we employed a DTM filtration with interval [0.03, 0.1] for all noise levels, following the practice of [2]. However, the scale of DTM-transformed data varies with the noise level, and the appropriate interval changes accordingly. We speculate that beyond $15\%$ noise, the adequate interval began to deviate from [0.03, 0.1]. The consequence of severe interval misspecification is more detrimental for ECC than for PH. In case of PLLay, a misspecified interval will result in the persistence landscape being represented as a zero vector, which is likely to be ignored during training. However, a misspecified interval causes ECC to contain unnecessary noise information, which can impede the model's training process. Hence, our insight is that the drop in performance results from the inherent instability of ECC coupled with optimization issues arising from inaccurately defined intervals for higher noise levels.

---

### Official Review · Reviewer_curU · 2024-11-03

**Soundness:** 2
**Presentation:** 2
**Contribution:** 3
**Rating:** 6
**Confidence:** 4

**Summary:**

The authors propose a novel learnable topological layer based on Euler Characteristic Curve (ECC). They prove that ECC is differentiable and approximate the gradient using distributional derivatives to be used at the time of back propagation. They show experimental results on some synthetic dataset and MNIST dataset.

**Strengths:**

The paper is well-written and easy to follow.

The authors are solving one of the important problems in the TDA in ML community of integrating topological descriptors in a learnable manner into ML pipelines.

The paper has a significant theoretical component.

**Weaknesses:**

Adding Table 4 and Table 5 from Appendix H.3 into the main text would help support the claim that though EC is not as expressive as PH, the performance is comparable.

However, I think that more extensive experimentation is required to understand the kind of data (graph data, point cloud data etc.) where trading off on the expressive power for computational efficiency does not result in a major decrease in performance.

The paper would also benefit from ablation studies on choosing the hyper parameters $v, T_{min}, T_{max} $ because these seem quite important to capture the relevant parts of a filtration.

**Questions:**

Looking at Table 4, would it be a right statement to make that on MNIST dataset, EC is able to capture the topology and generalize well only by looking at, say 100 or 200 examples? As the size of the training set grows, the performance gains start reducing. In which case, do you think it would be better for topological methods to be used when training data available is very limited?

Perslay and PLlay have options for permutation invariant functions. Any specific reasons to use the choices that you have? I find it quite surprising that EC consistently performs better than PH.

---

> ### Author Response · Authors · 2024-11-25
>
> We appreciate your valuable input and insights. We address each of your comments below.
>
> 1. **[Additional experiments]** Thank you for your suggestion. We completely agree that additional experiments on other data modalities would strengthen our work. In our future revised paper, we shall conduct additional experiments on the ORBIT5K point cloud data, a dataset that is often used as a benchmark in TDA.
>
> 2. **[Ablation study]:** We appreciate your suggestion on this matter. In response, we conducted an ablation study and included the results in Appendix I.5 of our revised paper.
>
> 3. **[Ans to Q1]:** We believe incorporating TDA features is most effective when the baseline model struggles to learn sufficient information from data. By intentionally restricting the size of data to 100 samples, we created a scenario where the baseline CNN encounters difficulties in learning adequate information. In such cases, the additional topological information provided by ECC notably improved model performance and generalization capabilities. Conversely, when the baseline model can effectively extract sufficient information from the data, the impact of incorporating TDA features may be minimal. Given the simplistic and macroscopic geometrical features of the MNIST dataset, the baseline CNN quickly captured enough information as the sample size increased, leading to a reduction in performance gain. When the dataset comprises more complex topological and geometric features that the baseline model struggles to capture, we conjecture that incorporating TDA features will lead to improvement in performance and generalizability, even as the training set size increases.
>
> 4. **[Ans to Q2]:** This is a good point. The MNIST dataset contains images of hand-written digits, each having at most 2 loop structures (e.g., digit 8 has 2 loops). To effectively capture these topological features, we selected the $top2$ function as the permutation invariant function for both models. Notably, the $top2$ function was also used in the MNIST experiment of the original PLLay paper[1] for the same purpose.
>
> Kwangho Kim, Jisu Kim, Manzil Zaheer, Joon Kim, Frédéric Chazal, and Larry Wasserman. "Pllay:Efficient topological layer based on persistent landscapes". [1]

---

> > ### Comment · Reviewer_curU · 2024-11-26
> >
> > Thank the authors for their efforts and their answers. I have read the answers and I would like to stick to my score.

---

### Official Review · Reviewer_qUCM · 2024-11-04

**Soundness:** 3
**Presentation:** 3
**Contribution:** 3
**Rating:** 5
**Confidence:** 3

**Summary:**

The paper introduces a neural network layer based on Euler Characteristic Curve (ECC), which is computationally much more efficient than layers based on Persistent Homology (PH). Compared to earlier work in this direction, this technique is shown to mitigate the vanishing gradient problem (Section 4).

**Strengths:**

The paper includes both theoretical results that guarantee the stability of the ECC layer (Section 5), and experiments that demonstrate its good performance (Section 6). It is also well written and organized.

**Weaknesses:**

See questions below.

**Questions:**

(Q1) Be mindful of the use of the word "topological" in the title, abstract and throughout the paper: Depending on the filtration, PH or ECC can capture geometric information too (which is e.g., the case for the Vietoris-Rips filtration too, that captures the size of the cycles). At least briefly comment on this.

(Q2) Somewhat related to (Q1) above, Turner et al. showed that ECC with respect to the height filtration from all directions describes the shape completely. Maybe this is relevant to mention too, and also comment why you don't opt for such an approach, as it is obviously more informative (more precisely, it provides complete information) than the limited aspects captured with ECC with respect to the filtrations you consider in your work?

(Q3) You write that the work most closely related to yours is by Kim and Röell. Why is PLLay more related to your approach than PersLay? What about ATOL [1]?

(Q4) When discussing related work, you write that the height filtration may be applicable to point clouds, yet with a compromise in connectivity information? Does this compromise need to happen? What if one uses non-vanilla Vietoris-Rips filtration with the height as the filtration function value on the vertices, and the geodesic/shortest-path distance as the filtration function on the edges? Also, does the work by Röell & Rieck not carry out experiments on MNIST and ModelNet point clouds?

(Q5) A (simplicial or) cubical complex is not a filtration (e.g. Section 3.1), we need a filtered cubical complex / filtration function value on each cube. I think you assume the filtration function to assign the greyscale pixel value, but this needs to be formulated more precisely. Other filtration functions are also reasonable, such as the DTM that you also use in this work, or height, density, dilation, erosion ..., see [2].

(Q6) More should be said about the choice of $T_{min}$ and $T_{max}$. For instance, for the example in Figure 1, choosing $T_{min} = 1$ and $T_{max} = 2$ would not give us any insights whatsoever? Can we not always take $T_{min}$ and $T_{max}$ to be respectively the minimum and maximum of the filtration function value across all simplices/cubes?

(Q7) Even more importantly, more can be said about the discretization of the interval $[T_{min}, T_{max}]$. The main advantage of PH is that one does not need to choose the filtration scale t that best captures the shape, rather, the shape is captured continuously across any value of t.  How do you know that there are no cycles that are both born and dead between some steps $t_i$ and $t_{i+1}$, and therefore not captured? This might be related to the assumptions of your crucial Proposition 5.1, but are never discussed. Also, if $\Delta t$ is very small (i.e., if one uses a very fine discretization to hopefully avoid the above issue), how useful is the very large constant on the right-hand side? This discussion is additionally important since you mention in Section 4.1 that the discretization is the problem for DECT by Röell, leading to gradient inconsistency.

(Q8) Can you visualize the different architectures (in the appendix)? Besides improved clarity, this might provide better insights on why ECC outperforms PH. Regarding this poor performance of PH in comparison to ECC, can you somewhat rely on Figure 5 (c) and (d), i.e., the behaviour of gradients, to provide some further intuition?

(Q9) Can you motivate the choice of the data sets in your experiments? Why not employ the data sets used in the paper that introduce your baselines such as PersLay, PLLay, DECT?

(Q10) In Section 6.1, you write that the time complexity is assessed by measuring the runtime for a complete iteration through the training data set. From Table 1, this means that PH is calculated for 60 000 28x28 images in only 33 seconds?

(Q11) Why do you say in Section 6.4 that the calculation of PH for Br35H 112 x 112 images is unfeasible, whereas it does work for the same data set, and even for 224 x 224 synthetic images in Section 6.1?

(Q12) Can you explain a bit better the "consequently" in the Discussion when you say that since ECC are topologically weaker invariants compared to PH, they exhibit less robustness? The latter is clear from the last paragraph of Section 5, but here you seem to imply that it is also expected? Intuitively, one would think that a signature is more stable if it captures less detailed information?

(Q13) Can you somewhat motivate the choice of hyperparameters? For instance, in Appendix H.4 you write that learning rates are 0.0001 and 0.01 respectively for ECLayr and ResNet?

[1] Royer, Martin, et al. "ATOL: Automatic topologically-oriented learning." arXiv preprint arXiv:1909.13472 (2019).

[2] Garin, Adélie, and Guillaume Tauzin. "A topological" reading" lesson: Classification of MNIST using TDA." 2019 18th IEEE International Conference On Machine Learning And Applications (ICMLA). IEEE, 2019.

I will raise my score if the main questions above (in particular Q7-Q9) are properly addressed.


Minor comments and typos:
- Explicitly mention that the proofs can be found in the Appendix (also to make it clearer that these results are not taken from the literature, but constitute the contribution of this paper).
- Stress more clearly the difference between C_X and E (ECC function and vector). For the sake of consistency, why not use E_X?
- A notation table (in appendix) could be helpful, especially when focusing on the main results, one needs to look through different parts of the paper to know what E, O, C are.
- Since you are comparing runtimes, I think it is useful to explicitly mention the size of the data sets and images in Section 6.
- References: Check the formatting, e.g., "Journal of Machine Learning Research" and "J. Mach. Learn. Res" are both used, the capitalization of journals is not consistent, "euler", etc.
- Appendix A: that is needed -> that are needed
- Appendix A, Footnote 1: reference is missing
- Appendix A: x. k -> x, k (2x)
- Appendix F title: Section -> Appendix. Also, should this Appendix not follow immediately after Appendix D, or at least after the two Appendices that contain the proofs for Section 4 and Section 5?
- Appendix 6: Proof of Proposition 5.2 -> Proof of Proposition 5.1

---

> ### Author Response · Authors · 2024-11-25
>
> We appreciate the great interest you have shown in our work, and address each of your questions below.
>
> 1. **[Ans to Q1]:** Thank you for pointing this out. We have addressed this in our revised paper, adding it as a footnote on page 3.
>
> 2. **[Ans to Q2]:** As the reviewer suggested, ECC with heights from all directions in Turner et al. has the full information of the shape. However, in the learning situation, having the full information is not directly transferred to the most useful learning in any sense (high accuracy, robust, etc). ECC with heights from all directions needs (with discretizing the directions) a high dimensional space to encode that information. Also, these encoded features are highly correlated, i.e., for e.g., when directions $v$ and $w$ are close, then corresponding ECC are very similar to each other as well. Hence in the setting where training data are not sufficient or moderate noise are present, ECC with all directions won't be appropriately trained due to its high dimensionality and correlation. Also, the height function from a direction is vulnerable to noise, in particular to outliers. With our approach, we can use a filtration that is robust to noise, for e.g., the distance to measure (DTM) filtration, and make the corresponding ECC robust to noise as well.
>
> 3. **[Ans to Q3]:** PLLay and DECT are most closely related works to ours, as they are the only prior works that address differentiability with respect to the layer input, enabling backpropagation through the layer. In contrast, PersLay introduces a framework for learning vectorizations of persistence diagrams within a deep learning context, but does not support backpropagation through the layer.
>
> 4. **[Ans to Q4]:** This is a good point. The compromise in connectivity information is discussed in the context of DECT [2], where the authors directly applied the height filtration to MNIST and ModelNet point clouds by simply counting the number of points above or below a hyperplane. While we find your suggested method plausible, it seems more aligned with applying the height filtration to a graph or mesh derived from a point cloud, rather than directly to the point cloud itself. Furthermore, the ECC from the non-vanilla Vietoris-Rips filtration as you suggested might be able to use the connectivity information fully, though it still suffers from noise, in particular to outliers, as we have explained in Ans to Q2.
>
> 5. **[Ans to Q5]:** Thank you for bringing this to our attention. We have addressed this in our revised paper to ensure clarity and avoid any potential confusion.
>
> 6. **[Ans to Q6]:** Yes, a naive and convenient way of setting the interval would be to assign $T_{min}$ and $T_{max}$ as the the minimum and maximum of the possible filtration values, respectively. Alternatively, one could select $[T_{min}, T_{max}]$ as a tighter interval within the range of possible filtration values, focusing on regions of the filtration that contain meaningful topological and geometrical information. Such an interval can be identified via hyperparameter search, or chosen intuitively by examining the ECC computed for some data. We have included a discussion on this topic in Appendix H of the revised manuscript.
>
> 7. **[Ans to Q7]** Thank you for this insightful question. As you pointed out, given the discretized interval, the vectorized ECCs are not able to capture cycles that are born and perished between $t_i$ and $t_{i+1}$. However, cycles with very short life spans are often insignificant (noisy) generators. Therefore, excessively dense discretization may not be beneficial, as including numerous small, noisy generators in the learning process can lead to increased variance. (please refer to the illustration provided in Figure 6 of the revised manuscript.) On the other hand, the discretization should also avoid being overly sparse, as this could compromise the ability to capture essential global features. Determining the optimal choice of discretizing bins is a non-trivial task, which we do not address in the current study. However, as demonstrated in our simulations, standard data-driven heuristics, such as cross-validation or a simple grid search, can be employed to select an appropriate value for $v$ that effectively balances the aforementioned trade-offs. Also importantly, avoiding excessively fine discretization is essential to prevent overly loose stability bounds in Proposition 5.1. We make this clear in our revised manuscript.

---

> ### Author Response · Authors · 2024-11-25
>
> 8. **[Ans to Q8]:** We absolutely agree that a visualization of the model architectures would enhance clarity and understanding of our work. Accordingly, we have included illustrations in Figure 8 of the revised manuscript. Regarding performance of ECC and PH, we speculate that the observed gap mainly arises from the inherent nature of the two descriptors, rather than differences in their gradients. For certain datasets, capturing the major macroscopic geometrical features is often sufficient for the classification task. In such cases, relying on more complex, high-resolution features can lead to overfitting. We further conjecture that the relative simplicity of ECC facilitates the optimization process compared to the more intricate features derived from PH, particularly in scenarios with limited data, thereby yielding improved performance. We note that, however, if the topological features at a more granular level are critical for the learning task, PH-based features are likely to yield superior performance. We will clarify this in the revised text.
>
> 9. **[Ans to Q9]:** Thanks for this question too. The datasets, many of which also have been explored in prior studies, are primarily used to demonstrate the advantages of applying TDA, as their underlying geometric structures provide valuable insights into the nature of the data. First, the classic MNIST dataset serves as an excellent example where prominent underlying geometric features (e.g., the number of holes, curvatures, etc.) can be effectively utilized for classification tasks. This dataset was also employed in [1] and [2], although the dataset was transformed into a point cloud/graph/mesh in [2]. The Br35H dataset, with its larger dimensions (112 × 112) compared to MNIST (28 × 28), features brain tumors that manifest as loops or connected components, making it particularly suitable for demonstrating ECC’s computational efficiency. Moreover, in our final revised paper, we shall conduct additional experiments on ORBIT5K point cloud data, a dataset that is often used as a benchmark in TDA and is also employed in [1] and [3].
>
> 10. **[Ans to Q10]** Yes, persistence diagrams for 60000 images of size 28x28 were calculated in 33 seconds on the Apple M2 cpu. We utilized the GUDHI package to compute PH, using a superlevel set filtration applied to a V-constructed cubical complex.
>
> 11. **[Ans to Q11]** We apologize for the error in our wording. Our intention was to convey that calculating PH for 112 x 112 images is ‘impractical’ or ‘inefficient’ due to computational burdens, rather than ‘infeasible.’ This has been corrected in the updated paper.
>
> 12. **[Ans to Q12]** ECC is considered a weaker topological invariant because it aggregates topological features across all homology dimensions, whereas PH provides separate topological summaries for each homology dimension. Intuitively, this makes ECC less stable than PH, as changes in the data can lead to more significant alterations in the aggregated information compared to the changes observed individually in each homology dimension. This intuition aligns with the mathematical proof: ECC is bounded by the Wasserstein distance, which is less strict than the Bottleneck distance that bounds PH. Moreover, even if we don't aggregate on homology dimensions, ECC is something like a rank function; all the noisy homological features at the given flitration value $t$ are accumulated to the rank function values at that $t$. Adaptively reflecting homological features according to their importances is almost impossible. While in the persistence diagram, noisy homological features are presented as individual points, and one can adaptively reflect homological features according to their importances: for e.g., using $\ell_{\infty}$ type distance such as the bottleneck distance will disregard noisy homological features below threshold. This phenomenon is also partially due to that persistence diagrams have much more complicated geometrical structure that can accommodate complicated topological difference, compared to functional spaces.

---

> ### Author Response · Authors · 2024-11-25
>
> 13. **[Ans to Q13]:** Our motivation was purely data-driven, as we conducted a hyperparameter search to identify the training hyperparameters that yield the best performance. Initially, we searched for the optimal training hyperparameters for the vanilla CNN/ResNet models without topological layers. Once these were determined, we attached the topological layers and performed a separate hyperparameter search for the topological layers, keeping the CNN/ResNet hyperparameters fixed to the previously identified optimal values.
>
> We hope the above response addresses your concerns. However, if there are any remaining issues, please feel free to let us know through your comments.
>
> [1] Kwangho Kim, Jisu Kim, Manzil Zaheer, Joon Kim, Frédéric Chazal, and Larry Wasserman. "Pllay:Efficient topological layer based on persistent landscapes".
>
> [2] Roell, Ernst, and Bastian Rieck. "Differentiable Euler characteristic transforms for shape classification".
>
> [3] Mathieu Carriére, Frédéric Chazal, Yuichi Ike, Théo Lacombe, Martin Royer, and Yuhei Umeda. "Perslay: A neural network layer for persistence diagrams and new graph topological signatures".

---

### Official Review · Reviewer_uRHy · 2024-11-04

**Soundness:** 3
**Presentation:** 2
**Contribution:** 3
**Rating:** 5
**Confidence:** 3

**Summary:**

This paper introduces ECLayr, a new topological layer for deep learning that uses the Euler Characteristic Curve (ECC) rather than traditional persistent homology. The key innovation is achieving efficient computation and stable backpropagation while still capturing useful topological features. The authors propose a new gradient approximation method that avoids the vanishing gradient issues found in previous approaches DECT using sigmoid approximations. They provide theoretical stability guarantees and demonstrate ECLayr's effectiveness through experiments on classification tasks with limited or noisy data, topological autoencoders, and high-dimensional medical imaging data.

**Strengths:**

- The computational efficiency gain compared to persistent homology approaches is significant - showing speed improvements of 10-30x while maintaining comparable performance.
- The theoretical foundation is solid, with careful stability analysis and proofs regarding the gradient approximation method.
- The empirical results are comprehensive, testing the method across different scenarios and comparing against relevant baselines.

**Weaknesses:**

- The empirical results, while interesting, are somewhat preliminary - the authors acknowledge that their simple architecture couldn't fully capture the nested relationship in the data.
- There could be more investigation into what topological features ECLayr is actually capturing and why they're useful for the specific tasks. The comparison to DECT could also be expanded, as it's the most similar existing approach.
- The hyperparameter selection process (like choice of filtrations and beta for gradient control) could be better explained.

**Questions:**

- How sensitive is the method to the choice of filtrations, the number of grid points and the intervals? Some ablation studies on these parameters would be informative.
- More insight into why ECLayr sometimes outperforms persistent homology-based methods despite capturing less topological information?

---

> ### Author Response · Authors · 2024-11-25
>
> Thank you for your time and valuable feedback. We would like to address each of your major concerns in the following.
>
> 1. **[Investigation on features captured by ECLayr]:** ECLayr captures the Euler characteristic of the underlying structure within the data. To facilitate understanding, we have included a visual illustration in Figure 6 of the updated manuscript. We hope this addition enhances the clarity of our work.
>
> 2. **[Presentation of hyperparameter selection process]:** Thank you for your valuable feedback. We fully agree that additional clarification on hyperparameter selection would strengthen our paper. To address this, we have included a new section in the Appendix of the amended manuscript, providing guidelines and insight on this topic. Below, we briefly summarize the key points. For a comprehensive explanation, including illustrations and examples, we kindly refer you to Appendix H of the revised paper.
>
> - Choice of filtration: The choice of filtration is fairly straightforward, as certain filtrations are commonly favored for specific data modalities and training contexts. Ultimately, the filtration that best captures the topological features of the given data should be selected.
>
> - Choice of $[T_{min}, T_{max}]$. A naive and convenient approach is to assign $T_{min}$ and $T_{max}$ as the minimum and maximum of possible filtration values, respectively. Alternatively, one could select $[T_{min}, T_{max}]$ as a tighter interval within the range of possible filtration values, focusing on regions of the filtration that contain meaningful topological and geometrical information. Such an interval can be identified via hyperparameter search, or chosen intuitively by examining the ECC computed for some data.
>
> - Choice of $v$: The spacing between grid points is important as our vectorized ECC does not account for cycles that are born and dead between $t_i$ and $t_{i+1}$, where $t_i, t_{i+1} \in tseq$. Provided that the discretization is not overly sparse, the uncaptured cycles are often small (noise) generators with life span shorter than $t_{i+1} - t_i$. This implies that with appropriate discretization, noise can be partially filtered by design. Therefore, using a highly dense discretization is not necessarily beneficial, as it captures even the small (noise) generators. Conversely, using a excessively sparse discretization may jeopardize the capturing of essential global features. The optimal choice of $v$ is not always evident; we recommend hyperparameter search using cross validation to determine the adequate $v$ that balances the two circumstances.
>
> - Choice of $\beta$: The optimal choice of $\beta$ is somewhat ambiguous. Unfortunately, we do not have a clear rationale for choosing $\beta$; it is contingent upon numerous factors, including model architecture and the specific task at hand. Therefore, we recommend conducting hyperparameter search via cross validation to select an appropriate $\beta$.
>
> 3.  **[Ans to Q1]:** We appreciate your suggestion on this matter. In response, we conducted an ablation study and included the results in Appendix I.5 of our revised paper.
>
> 4. **[Ans to Q2]:** This is because, for certain datasets, capturing the major macroscopic geometrical features is often sufficient for the classification task. In such cases, relying on more complex, high-resolution features can lead to overfitting. We further conjecture that the relative simplicity of ECC facilitates the optimization process compared to the more intricate features derived from PH, particularly in scenarios with limited data, thereby yielding improved performance. We note that, however, if the topological features at a more granular level are critical for the learning task, PH-based features are likely to yield superior performance. We will clarify this in the revised text.
>
> We hope the above response addresses your concerns. However, if there are any remaining issues, please feel free to let us know through your comments.

---

> > ### Comment · Reviewer_uRHy · 2024-12-03
> >
> > Thanks for the response.
> >
> > There are some follow up questions.
> >
> > Regarding the choice of filtration, the author mentioned different options of filtrations on different types of data. However,
> > is it possible that "useful" topological features can be different even for the same type of datasets? Since the ECLayr depends on the choice of filtration, the question is that, is there a way to choose or construct suitable filtrations for different datasets with same types? Or, could the filtration be parameterized and learned as a part of the whole model?

---

> > > ### Author Response · Authors · 2024-12-04
> > >
> > > Thank you for your insightful questions. We address each of them below.
> > >
> > > - A filtration can be thought of as a lens that provides a means of examining the data. Thus, it is possible that even for the same data type, "useful" topological features can vary based on the filtration used. For instance, the DTM filtration employed in our paper can focus on either local or global topological features depending on the choice of hyperparameter $m_0$.
> > >
> > > - The selection of suitable filtrations for different datasets of the same type is more intuitive than strictly rule-based. One must first decide the topological features to extract from the specific data, and subsequently choose the most appropriate filtration for capturing those features.
> > >
> > > - Parameterizing and learning the filtration is, to our knowledge, uncommon. Nonetheless, we believe it could be an interesting idea for future research.

---

### Author Response · Authors · 2024-11-25

We sincerely thank all the reviewers for their insightful questions and thoughtful comments. Based on your valuable feedback, we have made the following updates in our revised manuscript.
- A discussion highlighting the improvement in time complexity compared to the sigmoid approximation (Section 4).
- Guidelines and insights to assist in selecting appropriate TDA hyperparameters (Appendix H).
- Visualization of topological features captured by ECC (Figure 6).
- Visualization of model architectures (Figure 8)
- Ablation study on hyperparameter influence (Appendix I.5)

The major changes are highlighted in our revised paper.

---

### Author Response · Authors · 2024-12-04

We express our gratitude to all the reviewers for the valuable discussions and feedback, which significantly contributed to improving our paper, and hope that our responses have sufficiently resolved all of your concerns.

---

### Meta-Review · Area_Chair_h4Br · 2024-12-16

**Metareview:**

This submission proposes a new layer based on the Euler Characteristic Transform (ECT), a method drawing from geometry and topology to  characterise point clouds, graphs, and simplicial complexes. In contrast to existing work leveraging the ECT, the proposed layer has the advantage of improved computational stability with respect to backpropagation. Moreover, the submission also proves novel stability properties of the layer and shows its utility in different scenarios, namely its use as a loss term and its use in image classification.

*Strengths:*

- Providing additional theoretical and empirical insights into using the ECT in practice
- Showcasing several applications and providing a brief contextualisation in terms of existing methods in computational topology (primarily through runtime comparisons)

*Weaknesses:*

- The experiments appear to be somewhat preliminary, in particular in light of existing methods like [PersLay](https://arxiv.org/abs/1904.09378), [PLLay](https://arxiv.org/abs/2002.02778), the work by [Hofer et al.](https://arxiv.org/abs/1707.04041), and [DECT](https://arxiv.org/abs/2310.07630). This makes a comparison impossible. In particular given the problems raised with gradient instability, a reader's expectation would be that an in-depth comparison with DECT be performed on the datasets discussed in said paper. This would also serve to provide better insights into the utility of the ECT.

- In a similar vein, a better comparison to state-of-the-art deep learning techniques would be required. Such a comparison can also take the form of a contextualisation by citing predictive performance indicators whenever appropriate, but as it stands now, the submission does not take such comparisons into account. As someone with a background in computational topology, I am raising this point because a paper introducing (or building upon) methods that are not yet established in the core machine learning community always runs the risk of not being widely-read since practitioners will not be able to place/compare the performance claims. As such, one of the tasks of a strong submission should be to state more clearly to what extent a proposed method is a useful addition to the toolkit of practitioners. Given the preliminary state of the experimental section, this is not yet done effectively here.

- Finally, the novelty of the method is somewhat unclear to me (and reviewers—see below). This could be a consequence of the missing comparisons with DECT and should be improved in a revision of the paper. Specifically, it should be made very clear how the work differs from prior work like DECT, which, unlike the characterisation in this work, appears to be also applicable to different data modalities and filtrations. I realise that novelty is a somewhat vague concept, and its not the determining factor for my overall recommendation.

In conclusion, upon reading the paper, related literature, and the discussions, I share the reservations by reviewers and suggest rejecting the paper for now. While I believe the idea and setup to be sound, it would nevertheless require another round of reviews to check a new revision of the paper. This assessment is primarily based on weaknesses in the experimental setup and a missing contextualisation and comparison to other methods, in particular other deep-learning methods and DECT on the same datasets.

I understand that this is not the desired outcome for the authors but I hope that they can use the feedback to further improve their manuscript.

**Additional Comments On Reviewer Discussion:**

Reviewers raised concerns about the experimental setup (`uRHy`, `qUCM`, `curU`), hyperparameter selection (`uRHy`, `qUCM`), discretisation strategies/ablations (`qUCM`, `curU`), the theoretical derivation (`98ug`) and novelty (`98ug`, `zMtu`, `uRHy`). During the discussion, the authors addressed several aspects, in particular providing more information about the hyperparameter selection process and the discretisation strategy. However, reviewers mentioned that additional changes to the manuscript would be required (`qUCM`, `98ug`) and one reviewer specifically mentioned that the changes shown by the authors during the rebuttal phase did not serve to alleviate concerns about novelty or the overall effectiveness of the method (`98ug`), which I understand to be related to the points raised in my meta-review concerning the comparison/contextualisation.

As outlined above, I based my review on primarily on the (perceived) weak/preliminary experimental setup. While the rebuttal phase was overall very productive, this submission would be a good candidate for a "major revision" in a journal. I strongly believe in the utility of the method and hope the authors find the comments raised during the discussion phase productive as well.

---

### Decision · Program_Chairs · 2025-01-22

Reject